



# Solutions for Thermally-driven Reactive Transport and Porosity Evolution in Geothermal Systems ("Reactive Lauwerier Problem")

Roi Roded[1], Einat Aharonov[2], Piotr Szymczak[3], Manolis Veveakis[1], Boaz Lazar[2], and Laura E. Dalton[1]

[1]Civil and Environmental Engineering, Duke University, Durham, NC, USA
[2]Institute of Earth Sciences, The Hebrew University, Jerusalem, Israel
[3]Institute of Theoretical Physics, Faculty of Physics, University of Warsaw, Warsaw, Poland

*Correspondence to*: Roi Roded (roi.roded@mail.huji.ac.il)

**Abstract.** Subsurface non-isothermal fluid injection is a ubiquitous scenario in energy and water resources applications, which can lead to geochemical disequilibrium and thermally-driven solubility changes and reactions. Depending on the nature of the solubility of a mineral, the thermal change can lead to either mineral dissolution or precipitation (due to undersaturation or supersaturation conditions). Here, by considering this thermo-hydro-chemical scenario and by calculating the temperature-dependent solubility using a non-isothermal solution (the so-called *Lauwerier solution*), thermally-driven reactive transport solutions are derived for a confined aquifer. The coupled solutions, hereafter termed the "Reactive Lauwerier Problem", are developed for axisymmetric and Cartesian symmetries, and additionally provide the porosity evolution in the aquifer. The solutions are then used to study two common cases: (I) hot $CO_2$-rich water injection into carbonate aquifer; and (II) hot silica-rich water injection, leading to mineral dissolution and precipitation, respectively. We discuss the timescales of such fluid-rock interactions and the changes in hydraulic system properties. The solutions and findings here contribute to the understanding and management of subsurface energy and water resources, like aquifer thermal energy storage (ATES), aquifer storage and recovery (ASR) and reinjection of used geothermal water. The solutions are also useful for developing and benchmarking complex coupled numerical codes.



# 1 Introduction

The recharge or injection of fluids under constrained physical and chemical states in geothermal systems and aquifers is a common phenomenon in both natural and applied systems (Phillips, 2009; Stauffer et al., 2014). In many instances, thermal changes within these systems can shift the system from a state of geochemical equilibrium to disequilibrium, and lead to chemical reactions over extensive

distances, determined by the variations in temperature. These perturbations result from the changes in the solubility of minerals in the groundwater, which can become supersaturated or undersaturated in response to thermal changes. These thermally-driven reactions cause progressive changes in the rock porosity and hydraulic properties, resulting from the accumulation, removal or replacement of solid minerals, often accompanied by volumetric changes (Phillips, 2009; Woods, 2015). Such processes are

responsible for the natural transformations of rocks encompassing a spectrum from diagenesis and metamorphism (Jamtveit and Yardley, 1996; Yardley et al., 2011) to the evolution of aquifers and reservoirs (Andre and Rajaram, 2005; Jones and Xiao, 2006), and even to melt migration in the Earth's mantle (Aharonov et al., 1995; Kelemen et al., 1995). In applied systems, these fluid-rock interactions can significantly impact the hydrothermal performance at the timescale of years (Huenges et al., 2013;

Pandey et al., 2018).

In different systems, depending on the natural solubility of the minerals, an increase in temperature, for instance, can either induce dissolution or precipitation. This is because mineral solubilities can either increase with temperature (*prograde solubility*) or decrease with it (*retrograde solubility*; Jamtveit & Yardley, 1996; Woods, 2015). In the most common scenario, when a hot saturated fluid cools down,

prograde mineral solubility leads to supersaturation in the aqueous solution, causing the precipitation of solid minerals. Conversely, for minerals exhibiting retrograde solubility, a cooling saturated fluid will become undersaturated, enabling it to dissolve the surrounding minerals. Flow and transport commonly influence the state of saturation by continuously introducing thermally-disequilibrated fluid, which subsequently becomes geochemically disequilibrated. This occurs because, in many cases, advection

serves as the dominant transport mechanism, characterized by a shorter timescale ($t_A$) compared to diffusive heat ($t_C$) or diffusive solute transport ($t_D$). These timescales are represented by $t_A = l_A/u$, $t_C =$



$l_C^2/\alpha_b$, and $t_D = l_D^2/D$ where $l_A$, $l_C$ and $l_D$ are characteristic length scales of advection, heat conduction, and ionic diffusion, respectively. Here, $u$ denotes the Darcy flux [L T$^{-1}$], while $\alpha_b$ and $D$ are the bulk thermal diffusivity and ionic diffusion coefficient, respectively. The ratio of these timescales defines the thermal Péclet number ($Pe_T = t_C/t_A$) and the solute Péclet number ($Pe_s = t_D/t_A$), which are used to characterize the transport regime in these systems. When $Pe_T$ and $Pe_s$ are high (i.e., >> 1), advective transport prevails (Ladd & Szymczak, 2021; Nield & Bejan, 2017; Roded, Aharonov, Holtzman, et al., 2020).

The overall integrated action of the mechanisms described above results in a coupled *Thermo-Hydro-Chemical* (THC) process (Huenges et al., 2013; Pandey et al., 2018; Phillips, 2009; Regenauer-Lieb et al., 2013). The tightly coupled feedbacks in such THC processes commonly render them highly nonlinear: fluid flow and diffusive heat and solute transport induce chemical reactions, which, in turn, alter the pore structure and its transport properties, leading to further feedback on flow and transport (Chaudhuri et al., 2013; Phillips, 2009; Woods, 2015). Over time, this process can give rise to the emergence of distinct porosity patterns in the system. This self-organization process depends on the flow and transport regime as well as both the initial and boundary conditions (Aharonov et al., 1997; Ortoleva et al., 1987a; Roded et al., 2021; Spiegelman et al., 2001; Szymczak and Ladd, 2009).

Studying these coupled feedback alterations improves the understanding of reactive transport processes taking place in the Earth's upper crust and geophysical properties. Specifically, these studies are integral to the sustainable planning and long-term management of water resources (Andre and Rajaram, 2005; Phillips, 2009), hydrocarbon recovery (Jones and Xiao, 2013; Lauwerier, 1955), operation of geothermal energy systems (on the scale of tens of years; Frick et al., 2011; Huenges et al., 2013; Pandey et al., 2018), and CO$_2$ geo-sequestration (Dávila et al., 2020; Steefel et al., 2013; Tutolo et al., 2015). Additionally, there is the possibility of combining CO$_2$ geo-sequestration and geothermal energy production in enhanced geothermal systems (EGS). EGS are based on circulating fluid (in this case CO$_2$) between injection and production wells (Esteves et al., 2019; Luo et al., 2014).



Particularly in EGS, channelized dissolution can create a short circuit and reduce the heat exchange between the rock and the fluid. Conversely, precipitation can significantly reduce permeability, leading to reduced production and potentially sealing of reservoirs (Huenges et al., 2013; Olasolo et al., 2016;

Pandey et al., 2018). This aspect holds crucial importance in the global transition toward greener energy platforms and the increasing utilization of geothermal applications. The sustainability of these systems relies heavily on our comprehension of underlying fluid-rock interactions (Frick et al., 2011; Glassley, 2014; Pandey et al., 2018). Another challenge associated with geothermal utilization is the risk of groundwater contamination, where thermal changes and fluid mixing can lead to the leaching of

undesired chemical species from the rocks. Specifically, contamination may arise from the reinjection of fluids required to maintain reservoir pressure, or from the growing deployment of Aquifer Thermal Energy Storage (ATES) systems, that commonly leverage seasonal temperature differences (Bonte et al., 2014; Glassley, 2014; Possemiers et al., 2014). Moreover, substantial injections of hotter or colder water are frequently part of groundwater management practices, e.g., Aquifer Storage and Recovery

(ASR), where surplus water is stored and later recovered (e.g., during dry seasons; Maliva, 2019; Zheng et al., 2021).

In terms of mineralogy, a range of thermally-driven reactions occurs in the above-mentioned systems. Commonly reported precipitates accumulating in geothermal plant piping loops and natural spring deposits include minerals such as carbonates (e.g., calcite, dolomite, and siderite), sulfates (e.g., gypsum

and barite), and amorphous silica (Glassley, 2014; Huenges et al., 2013). Geothermal systems composed of sandstones and carbonates (i.e., limestone and dolomite) are ubiquitous globally in the crust and are prone to alterations (Goldscheider et al., 2010; Pandey et al., 2018; Wood and Hewett, 1984). The solubility of silica is proportional to temperature (i.e., prograde solubility), and water pumping or injection can lead to dissolution and/or precipitation, causing substantial changes in

reservoir transmissivity over time that can affect heat extraction (Pandey et al., 2018; Rawal and Ghassemi, 2014; Taron and Elsworth, 2009). In particular, silica precipitation can occur relatively fast, typically by several orders of magnitude faster compared to dissolution of either rocks of quartz minerals or amorphous silica (Rimstidt and Barnes, 1980). The exception is the dissolution of unconsolidated amorphous silica sediments (e.g., diatomite), which due to their very high specific





reactive surface area can be intensely dissolved when steam and hot water, undersaturated with respect to silica are injected (Bhat and Kovscek, 1998).

In contrast to silica (and most rock-forming minerals), carbonate minerals demonstrate an inverse relation (i.e., retrograde solubility), which is often strong and influenced by $CO_2$ content. Consequently, limestone and dolomite aquifers and reservoirs subjected to geothermal flows, commonly rich in $CO_2$,

can evolve at relatively short timescales (Andre and Rajaram, 2005; Coudrain-Ribstein et al., 1998; Roded et al., 2023). Either rapid dissolution or rapid precipitation can occur in such systems, depending on conditions: precipitation can be induced by heating and/or due to $CO_2$ degassing resulting from a decrease of groundwater pressure, often leading to the formation of various deposits in springs and caves (known as "speleothems"; Ford and Williams, 2013; Jamtveit and Hammer, 2012). Conversely,

aggressive karst formation processes are induced by the cooling of deep-origin (> 1 km) $CO_2$-rich thermal fluids that upwell to shallower depths (so-called *hypogene karst*; Chaudhuri et al., 2013; Jones & Xiao, 2006, 2013; Roded et al., 2023). If these THC dissolution processes are localized, kilometer-long cave systems can develop over relatively short geological timescales (tens of thousands of years; Roded et al., 2023). In fact, due to the intense fluid-rock interactions, carbonate layers may not be

appropriate candidates for EGS technology (Pandey et al., 2018).

Investigating the multi-physical systems of thermally-controlled reactive flow is complex and relies largely on numerical models, which are facilitated by the ongoing advancements in computational and algorithmic capabilities (Kolditz et al., 2016; Pandey et al., 2018; Steefel et al., 2015). This allows researchers to develop numerical models of an ever-increasing complexity, accounting for geophysical

scenarios of complex flow, mass, and heat transport with multiple chemical species and reactions across different settings. Over recent decades, these models have significantly improved our understanding of subsurface processes (Niemi et al., 2017; Regenauer-Lieb et al., 2013; Seigneur et al., 2019; Steefel et al., 2013). However, the validity of such models remains questionable if the results cannot be rigorously tested against field observations, laboratory results, and analytical solutions (Kolditz et al., 2016; Nield

and Bejan, 2017). Analytical solutions, in particular, allow the establishment of functional relationships





between variables and physical properties, providing robust reliability and accuracy tests for numerical models (Diersch and Kolditz, 2002; Nield and Bejan, 2017; Bear and Cheng, 2010).

However, testing multi-coupled THC codes to a satisfactory level is often mathematically cumbersome and, as such, precluded by many approaches. The limitation arises because existing theoretical solutions focus solely on scenarios related to heat and/or solute transport (Diersch & Kolditz, 2002; Nield & Bejan, 2017; Stauffer et al., 2014; Turcotte & Schubert, 2002 and the references therein), or reactive solute transport (Bear and Cheng, 2010; Nield and Bejan, 2017; and the references therein), and complete solutions coupling THC processes are scarce (White et al., 2018). In fact, to the best of our knowledge, coupled THC solutions are limited to only two scenarios: thermally-driven reactive front development (Jupp and Woods, 2003, 2004), and cases involving thermal and/or solutal convection in a reactive medium (such as the Rayleigh–Bénard equivalent in a reactive porous medium; Al-Sulaimi, 2015; Corson & Pritchard, 2017). However, solutions for fundamental and practical situations in geothermal and groundwater systems, such as non-isothermal injection into a reservoir and consequent matrix modifications, are missing. This is despite the existence of the so-called *Lauwerier solution* (Lauwerier, 1955), which analytically predicts the thermal field resulting from hot (or cold) fluid injection into a thin non-reactive confined layer system. The Lauwerier solution served as the basis for the development of an increasing number of different modified heat transport solutions, accounting for various boundary conditions and system geometries, considering conduction and dispersion, and even accommodating fractured media (Abbasi et al., 2017; Chen & Reddell, 1983; Lin et al., 2019; Shaw-Yang & Hund-Der, 2008; Voigt & Haefner, 1987; Yang et al., 2010; Zhou et al., 2019; Ziagos & Blackwell, 1986; see review in Stauffer et al., 2014).

This work presents analytical and numerical solutions, invoking non-isothermal fluid injection from a point or planar source into a thin confined aquifer (essentially the same scenario as of the aforementioned Lauwerier problem). However, in this study, thermal changes drive reactions and porosity evolution. In fact, here we define and solve the coupled physics of the *reactive Lauwerier problem*. To achieve this, we employ a temperature-dependent solubility in a reactive-flow formulation, while accounting for the thermal field following the Lauwerier formulation. The equations are

subsequently solved for radial and planar flows in the aquifer. Next, the general solution is applied to two ubiquitous scenarios: carbonate aquifer dissolution and silica precipitation in the aquifer, along with
the respective permeability evolutions of each aquifer.

## 2 Mathematical Analyses

### 2.1 Reactive Lauwerier Scenario and the Conceptual Model

We consider Lauwerier problem settings (Lauwerier, 1955; Stauffer et al., 2014) involving the injection of hot (or cold) fluid into a confined aquifer located between bedrock and caprock with lateral flow
along the coordinate, $\varphi$. The latter can represent the radial coordinate in an axisymmetric setting or $x$ in Cartesian coordinates, i.e., $\varphi = r$ or $x$. Figure 1 illustrates a summary of the problem, while Table 1 provides a summary of the nomenclature.

solubility, $c_s(T)$, and hence disequilibrium and reaction, which in turn drives evolution of the porosity of the aquifer from its initial value, $\theta_0$. $z$ represents the vertical coordinate. In the main text both polar and Cartesian
geometries are considered, with $\varphi = r$ or $x$, respectively. The origin of $\varphi$ and $z$ is defined at the center of the injection well. The injection well exhibits either axial (as shown in the sketch) or planar symmetry if Cartesian geometry is considered.

Downstream, along the flow path away from the injection point, heat is exchanged between the aquifer and the impermeable confining rock layers. Within the confining layers, heat is transported by
conduction alone. The heat exchange and thermal variations in the aquifer induce changes in the solubility of the minerals (i.e., saturation concentration, $c_s(T)$), which in turn trigger undersaturation and dissolution reactions, or conversely, supersaturation and precipitation reactions that modify the aquifer porosity, $\theta$. Both the removal or accumulation of minerals can occur, depending on the injection temperature (colder or warmer than ambient) and the prograde or retrograde nature of the reactive
minerals. Our radial setup pertains to injection from a single well or mimics natural localized thermal upwelling in fractured/faulted media of deep-origin, discharging into the shallower aquifer (Craw, 2000; Micklethwaite and Cox, 2006; Roded et al., 2013, 2023; Tripp and Vearncombe, 2004). The planar source setup simulates injection wells arranged in a straight row (Lauwerier, 1955).





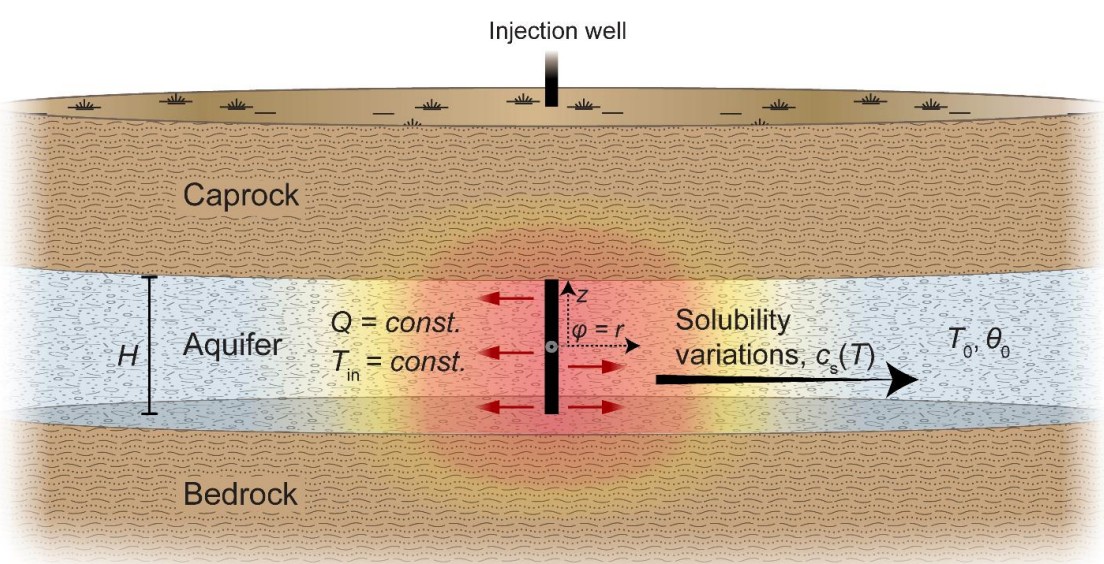


**Figure 1: Sketch of the reactive Lauwerier problem and the conceptual model for thermally-driven reactive transport in geothermal systems (the radial case).** Hot (or cold) fluid is injected into a confined aquifer between aquiclude bedrock and caprock at a constant flow rate, $Q$, and temperature, $T_{in}$. The initial temperature of the aquifer is $T_0$ and its thickness is $H$. Downstream, along the flow path, heat is conducted from the aquifer through the confining layers. Thermal variations in the aquifer (color gradients) induce changes in

## 2.2 Main Model Assumptions

Here, the THC conceptual model of Fig. 1 is described mathematically using conservation equations for heat and reactive transport along with initial and boundary conditions. The thermal Lauwerier solution and the mathematical model involve several simplifying assumptions, the major ones of which are listed below. For a more comprehensive overview, expanded versions of the conservation equations are provided in Appendix A.

The underlying thermal assumptions include negligible basal (background) geothermal heat flow and an initial geothermal gradient compared to the heat input by the injected fluid. The aquifer is located at a significant depth preventing heat transport to the surface, otherwise, greater heat exchange would occur between the aquifer and the caprock. This assumption regarding the depth also depends on the timescale




of interest: the thermal front, which ascends with time, may not reach the surface on a short timescale.

However, it may transport heat to the surface after a longer time (which can be estimated using $t_C$).

| Table 1. Nomenclature | | | |
|---|---|---|---|
| **Roman** | | $T$ | Temperature, °C |
| $A_s$ | Specific reactive surface area, m$^2$/m$^3$ | $u$ | Fluid velocity, m/s |
| $c$ | Solute concentration, mol/m$^3$ | $x$ | Coordinate, m |
| $c_s$ | Saturation concentration, mol/m$^3$ | $y$ | Coordinate, m |
| $c_{sol}$ | Concentration of soluble solid, mol/m$^3$ | $z$ | Coordinate, m |
| $C_p$ | Volumetric heat capacity, J/(m$^3$ °C) | **Greek** | |
| $D$ | Diffusion coefficient, m$^2$/s | $\alpha$ | Thermal diffusivity, m$^2$/s |
| $Da$ | Damköhler number | $\beta$ | Solubility change parameter, mol/(m$^3$ °C) |
| erf | Error function | $\gamma$ | Acid capacity number |
| erfc | Complementary error function | $\delta\theta$ | Small change of porosity |
| $Err$ | Error | $\Delta$ | Total difference |
| $H$ | Aquifer thickness, m | $\eta$ | Parameter group, m$^{-2}$ |
| $k$ | Permeability, m$^2$ | $\theta$ | Porosity |
| $k_{eff}$ | Effective permeability, m$^2$ | $\Theta$ | Heat exchange term, W/m$^2$ |
| $K$ | Thermal conductivity, W/(m °C) | $\lambda$ | Reaction rate coefficient, m/s |
| $l$ | Characteristic length scale, m | $\Lambda$ | Solute disequilibrium, mol/m$^3$ |
| $l_A$ | Characteristic length scale of advection, m | $\mu$ | Fluid viscosity, Pa s |
| $l_C$ | Characteristic length scale of conduction, m | $v$ | Stoichiometric coefficient |
| $l_D$ | Characteristic length scale of diffusion, m | $\zeta$ | Parameter group, m$^{-2}$ |
| $n$ | Exponent of $\theta$-$k$ relation | $\rho$ | Density, kg/m$^3$ |
| $p$ | Fluid pressure, Pa | $\sigma$ | Parameter group, m$^{-1}$ |
| $Pe_s$ | Solute Péclet number | $\varphi$ | Lateral coordinate, $\varphi = r$ or $x$, m |
| $Pe_T$ | Thermal Péclet number | $\omega$ | Parameter group, m$^{-1}$ |
| $Q$ | Total volumetric flow rate, m$^3$/s | $\Omega$ | Reaction rate, mol/(m$^3$ s) |
| $r$ | Coordinate, m | **Subscripts** | |
| $R$ | Effective permeability radius, m | Apr | Approximated value |
| $t$ | Time, s | b | Bulk rock |
| $t_A$ | Characteristic timescale of advection, s | Ext | Exact value |
| $t_C$ | Characteristic timescale of conduction, s | f | Fluid |
| $t_D$ | Characteristic timescale of diffusion, s | in | Inlet |
| $t_M$ | Characteristic timescale of mineral alteration, s | max | Max |
| $t'$ | Time parameter, s | 0 | Initial average quantity |



Heat transport in the layers confining the aquifer is described by conduction, and only in the vertical direction ($z$), neglecting lateral ($\varphi$) heat conduction. This assumption limits the applicability of the solution to scenarios involving large, injected fluid fluxes. To assess the validity of this assumption, a thermal Péclet number, which compares heat advection in the aquifer to lateral heat conduction, $Pe_T = ul/\alpha_b$, is used. $Pe_T$ involves a length scale, $l$, at which substantial temperature variation occurs (e.g., larger than 2 % from the total temperature change, $\Delta T$). Analysis using the parameter values from Table 2 and the results of section 3 (i.e., *a posteriori* inspection) confirms $Pe_T \gg 1$ at all times. Additionally, beyond very early moments, the length scale $l$ should be larger than the vertical dimension of the aquifer, $H$, at which complete thermal mixing is assumed. This assumption may not be applicable if a thick aquifer (i.e., large $H$) is considered and substantial vertical temperature gradients are expected to develop.

Furthermore, conduction and solute diffusion within the aquifer groundwater is neglected because the respective thermal ($Pe_T$) and solute ($Pe_s$) Péclet numbers are assumed to be large. Fluid and solid properties, such as density and heat conductivity, are considered constant and independent of temperature. Also, the specific reactive surface area, $A_s$, ($L^2$ to $L^{-3}$ of porous medium) is considered constant here and assumed not to change as reaction progresses. In most instances, this assumption does not weaken the applicability of the solution, since $A_s$ may vary widely across different rock lithologies, e.g., from $10^{-1}$ m$^{-1}$ in fractured media (Deng and Spycher, 2019; Pacheco and Van der Weijden, 2014) to above $10^5$ m$^{-1}$ for porous rocks (Mostaghimi et al., 2013; Noiriel et al., 2012; Seigneur et al., 2019) and can often only be estimated very roughly (e.g., within an order of magnitude accuracy). Furthermore, $A_s$ can evolve with the reactive flow in a way that is difficult to estimate (Noiriel, 2015; Seigneur et al., 2019). However, if large porosity changes are considered, the inherent assumption of constant $A_s$ can limit the applicability of the solutions.

## 2.3 The Basic Conservation Equations

Neglecting heat conduction in the radial direction, $r$, the heat conduction equation in the rock confining the aquifer above and below is given by:



$$\frac{\partial T}{\partial t} = \alpha_b \frac{\partial^2 T}{\partial z^2}, \qquad \begin{cases} z \leq -\dfrac{H}{2} \\ z \geq \dfrac{H}{2} \end{cases}, \tag{1}$$

where $T$ represents temperature, $t$ denotes time, $z$ is the vertical coordinate with its origin at the center of the injection well and $H$ is the aquifer thickness (see Fig. 1). The quantity $\alpha_b = K_b/C_{p_b}$ is the thermal diffusivity [$L^2 T^{-1}$], where the subscript b indicates bulk rock, $K$ is the thermal conductivity, and $C_p$ is the volumetric heat capacity (Chen and Reddell, 1983; Stauffer et al., 2014).

Assuming that heat transport in the fluid along the aquifer is governed by advection and that complete
mixing occurs in the aquifer transverse direction ($z$), a "depth-averaged" heat-transport equation can then be formulated for the aquifer region:

$$C_{p_b} H \frac{\partial T}{\partial t} = -C_{p_f} H \frac{1}{r} \frac{\partial (ruT)}{\partial r} - \Theta(r,t), \quad \text{for} \quad -\frac{H}{2} \leq z \leq \frac{H}{2}, \tag{2}$$

where subscript f denotes fluid and $u(r)$ is the fluid velocity (or Darcy flux), which can be determined from the total volumetric flow rate, $Q$, using $u = Q/(H2\pi r)$ (assuming $u$ to be uniform along the $z$
direction of the aquifer; Andre & Rajaram, 2005; Lauwerier, 1955). The function $\Theta$ accounts for the heat exchange between the aquifer and the confining rock located above and below, calculated using Fourier's law with continuous temperature assumed at the interfaces:

$$\Theta = -2K_b \frac{\partial T}{\partial z}\bigg|_{z=\frac{H}{2}, -\frac{H}{2}}. \tag{3}$$

The factor of two accounts for the rock both above and below the horizon (Stauffer et al., 2014).

The solute transport advection-reaction equation in the aquifer is:

$$0 = -u \frac{\partial c}{\partial r} - \Omega(r,t), \quad \text{for} \quad -\frac{H}{2} \leq z \leq \frac{H}{2}. \tag{4}$$

Here $c$ is the solute concentration [$M/L^3$] and $\Omega$ is the reaction term (Chaudhuri et al., 2013; Szymczak and Ladd, 2012). Eq. 4 is derived by neglecting transient and diffusive terms in the advection-diffusion-reaction equation (Eq. A.3 in Appendix A). The justification for the quasi-static approximation used in



deriving Eq. 4, lies in the separation of timescales between heat conduction ($t_C$) in the confining rocks and mineral alteration ($t_M$), and the relaxation of solute concentration ($t_A$) (for in-depth analysis and discussion see Appendix B and e.g., Detwiler & Rajaram, 2007; Ladd & Szymczak, 2017; Lichtner, 1991; Roded, Aharonov, Holtzman, et al., 2020; Sanford & Konikow, 1989).

Here, we assume surface-controlled reaction and first-order kinetics

$$\Omega = A_s \lambda \Lambda, \tag{5}$$

where $A_s$ is the specific reactive surface area of the reacting mineral ($L^2$ to $L^{-3}$ of porous medium) and $\lambda$ is the kinetic reaction rate coefficient [L T$^{-1}$], here assumed constant (Dreybrodt et al., 2005; Seigneur et al., 2019). $\Lambda$ is denoted as the solute disequilibrium and is defined as the difference between the concentration of dissolved ions and saturation (equilibrium) concentrations, $c_s$,

$$\Lambda = c - c_s(T). \tag{6}$$

Thus, the solute disequilibrium, $\Lambda$, is positive for undersaturation and negative for supersaturation. $c_s$ is calculated as:

$$c_s(T) = c_s(T_0) + \beta(T - T_0). \tag{7}$$

Here, $T_0$ represents the initial temperature in the aquifer and the parameter $\beta = \partial c_s / \partial T$. Eq. 7 assumes a

linear relationship between $c_s$ and $T$, with a constant proportionality factor $\beta$, which is positive for minerals of prograde solubility and negative for minerals of retrograde solubility (Al-Sulaimi, 2015; Corson and Pritchard, 2017; Woods, 2015).

Given the reaction rate (Eq. 5), the change in porosity, $\theta$, can be calculated as:

$$\frac{\partial \theta}{\partial t} = -\frac{\Omega}{\nu c_{sol}}, \quad \text{for} \quad -\frac{H}{2} \leq z \leq \frac{H}{2}, \tag{8}$$





where $c_{\mathrm{sol}}$ is the concentration of soluble solid mineral and $v$ accounts for the stoichiometry of the

reaction. In the case of planar flow and Cartesian coordinates, $r$ can be replaced by $x$ in the equations

above, while Eq. 2 takes the following form,

$$C_{\mathrm{p_b}}\frac{\partial(HT)}{\partial t} = -uC_{\mathrm{p_f}}H\frac{\partial T}{\partial x} - \Theta(x,t), \quad \text{for} \quad -\frac{H}{2} \leq z \leq \frac{H}{2}. \tag{9}$$

**2.4 Initial and Boundary Conditions**

The initial conditions involve a uniform temperature $T_0$ throughout the medium. The boundary

conditions at the injection well ($\varphi = 0$) include a constant rate of fluid injection at temperature $T_{\mathrm{in}}$ and

initially zero solute disequilibrium, $\varLambda = 0$ (Eq. 6). The caprock and bedrock thickness and aquifer extent

are assumed to be infinite.

**2.5 Solution of the Reactive Lauwerier problem**

**2.5.1 Axisymmetric (Radial) Flow**

**Aquifer temperature.** The solution of Eqs. 1 and 2 for the temperature distribution in the aquifer

(known as the Lauwerier solution) for the radial case is given by:

$$T(r,t) = T_0 + \Delta T \operatorname{erfc}[\zeta(r,t)r^2]. \tag{10}$$

Here, erfc is the complementary error function, $\Delta T = T_{\mathrm{in}} - T_0$ is the difference between injection and

initial aquifer temperature, and $\zeta$ is defined as:

$$\zeta(r,t) = \frac{\pi\sqrt{K_{\mathrm{b}}C_{\mathrm{p_b}}}}{QC_{\mathrm{p_f}}\sqrt{t'}}. \tag{11}$$

The time variable $t' = t - 2rC_{\mathrm{p_b}}/(C_{\mathrm{p_f}}u)$, and the solution given by Eq. 10 holds when $t' > 0$ (Stauffer et

al., 2014). We additionally assume long enough time and conditions where $t' \approx t$ (see Appendix C for

analysis of the validity of this assumption). Furthermore, for simplicity, we assume equal heat capacities

for both the confining rocks and the aquifer.





**Reactive solute transport.** We begin by substituting Eq. 6 into 4 to obtain:

$$0 = -u\left(\frac{\partial \Lambda}{\partial r} + \frac{\partial c_s}{\partial r}\right) + \Omega. \tag{12}$$

The derivative $\partial c_s/\partial r$ can then be expressed by differentiating the relationship in Eq. 7,

$$\frac{\partial c_s}{\partial r} = \frac{-\beta \partial T}{\partial r}, \tag{13}$$

and further substituting Lauwerier solution (Eq. 10), which provides:

$$\frac{-\beta \partial T}{\partial r} = 4\Delta T \frac{\beta \zeta r}{\sqrt{\pi}} e^{\left(-\zeta^2 r^4\right)}. \tag{14}$$

Next, substituting Eq. 14 into Eqs. 13 and 12 results in a linear inhomogeneous differential equation. Assuming saturation conditions at the inlet and the boundary condition of $\Lambda(r=0) = 0$, leads to the solution

$$\Lambda = \Delta T \beta e^{\left(\frac{\eta^2}{4\zeta^2} - \eta r^2\right)} \left(\text{erf}\left[\zeta r^2 - \frac{\eta}{2\zeta}\right] + \text{erf}\left[\frac{\eta}{2\zeta}\right]\right), \tag{15}$$

where erf is the error function and $\eta = H\pi A_s \lambda/Q$. Appendix D presents an approximation for Eq. 15 which is useful for efficient computation and prevents integer overflow (Press et al., 2007). Given the reaction rate (Eq. 5), the erosion and porosity change can be calculated based on the solid erosion equation

$$\frac{\partial \theta}{\partial t} = -\frac{\Omega}{\nu c_{\text{sol}}}, \tag{16}$$





where $c_{\text{sol}}$ is the concentration of soluble solid material and $v$ accounts for the stoichiometry of the reaction. Substituting Eq. 15 into Eq. 16, integrating over time, and using the initial condition of $\theta(t=0) = \theta_0$, results in a closed-form expression for the temporal and spatial evolution of porosity, $\theta$,

$$\theta(r,t) = \theta_0 + 4\frac{\zeta^2 t}{\eta^2}\frac{\lambda A_s \Delta T \beta}{v c_{\text{sol}}}\left(-e^{\eta/4\left(\frac{\eta}{\zeta^2}-4r^2\right)}\left(\text{erf}\left[\zeta r^2 - \frac{\eta}{2\zeta}\right]+\text{erf}\left[\frac{\eta}{2\zeta}\right]\right)+\frac{\eta}{\zeta\sqrt{\pi}}e^{-\eta r^2}\right.$$
$$\left.+\text{erf}[\zeta r^2](1-\eta r^2)-\frac{\eta}{\zeta\sqrt{\pi}}e^{-\zeta^2 r^4}+\eta r^2 - 1\right). \tag{17}$$

### 2.5.2 Planar Flow

In the Cartesian case, with injection along a line, the Lauwerier solution is,

$$T(x,t) = T_0 + \Delta T \text{erfc}[\omega(x,t)x], \tag{18}$$

where $\omega$ is defined as:

$$\omega(x,t) = \frac{\sqrt{K_b C_{p_b}}}{H C_{p_f} u \sqrt{t'}}, \tag{19}$$

and $t' = t - x C_{\text{pb}}/(C_{\text{pf}}u)$. Similarly, to the radial case, the solution holds at sufficiently long times, for which $t' \approx t$. Following the analogous steps as in the radial case, the solution is derived as:

$$\Lambda = \Delta T \beta e^{\left(\frac{\sigma^2}{4\omega^2}-\sigma x\right)}\left(\text{erf}\left[\omega x - \frac{\sigma}{2\omega}\right]+\text{erf}\left[\frac{\sigma}{2\omega}\right]\right), \tag{20}$$

and

$$\theta(x,t) = \theta_0 + 4\frac{\omega^2 t}{\sigma^2}\frac{\lambda A_s \Delta T \beta}{v c_{\text{sol}}}\left(-e^{\sigma/4\left(\frac{\sigma}{\omega^2}-4x\right)}\left(\text{erf}\left[\omega x - \frac{\sigma}{2\omega}\right]+\text{erf}\left[\frac{\sigma}{2\omega}\right]\right)+\frac{\sigma}{\omega\sqrt{\pi}}e^{-\sigma x}\right.$$
$$\left.+\text{erf}[\omega x](1-\sigma x)-\frac{\sigma}{\omega\sqrt{\pi}}e^{-\omega^2 x^2}+\sigma x - 1\right), \tag{21}$$

where $\sigma = A_s\lambda/u$.





## 3 Thermally-driven Reactive Flow in Geothermal Systems

In this section, we use the radial solutions presented in previous section, to examine two common scenarios: (I) injection of $CO_2$-rich hot water into a carbonate aquifer and (II) injection of silica-rich hot water into a sandstone aquifer. These scenarios result in cooling-induced calcite dissolution and silica

precipitation, respectively. The subsequent changes in porosity within these systems are then used to estimate the evolution of aquifer permeability. These scenarios are pertinent, for instance, in aquifer thermal storage, reinjection of geothermal water at shallow depths, or applications of groundwater storage and recovery (Diaz et al., 2016; Fleuchaus et al., 2018; Maliva, 2019).

### 3.1 Aquifer Properties and Injection Conditions

Here, we discuss conditions for thermally-induced reactivity in carbonates and sandstone aquifers and the parameter values assigned in the simulations (Table 2). Regarding the description of the kinetics of these systems, calcite dissolution can often be complex, involving various chemical species and reactions of varying orders (Dreybrodt, 1988; Plummer et al., 1978). However, for a wide range of pH values, it can be simplified and described by assuming a linear dependence on undersaturation or acid

concentration. Specifically, first-order kinetics are commonly employed to study natural karst formations (pH ~ 6; Dreybrodt et al., 2005; Palmer, 1991), dissolution under the acidic conditions common in engineering applications (Hoefner and Fogler, 1988; Peng et al., 2015), or in geothermal systems of high $CO_2$ partial pressure, $PCO_2$ (pH ~ 3; Coudrain-Ribstein et al., 1998; Lu et al., 2020; Roded et al., 2023). Silica precipitation can be well described by first-order kinetics (Carroll et al.,

1998; Ji et al., 2023; Pandey et al., 2015; Rimstidt and Barnes, 1980).

We also exploit approximately linear temperature-solubility dependence over the temperature range studied here (between $T_0 = 20\ °C$ and $T_{in} = 60\ °C$) and assign a constant $\beta$ value (Eq. 7; Andre and Rajaram, 2005; Glassley, 2014; Rimstidt and Barnes, 1980; Roded et al., 2023). Additionally, it should be noted that in carbonates, the temperature-solubility relation strongly depends on $PCO_2$: higher $PCO_2$

values result in larger increases in $c_s$ as the water cools (i.e., the magnitude of $\beta$ is larger, see figure 2b in Roded et al., (2023) and Andre & Rajaram, (2005); Palmer, (1991)). Here, in accordance with typical





conditions in geothermal systems, we consider injection of water with $PCO_2$ = 0.03 MPa (Coudrain-Ribstein et al., 1998; Lu et al., 2020).

| Table 2. Parameter values used in the simulations. | |
|---|---|
| Aquifer thickness | $H = 4$ m |
| Initial porosity | $\theta_0 = 0.05$ and $0.2$ |
| Total volumetric flow rate[1] | $Q = 500$ m³/s |
| Initial aquifer temperature[2] | $T_0 = 20$ °C |
| Injection temperature[2] | $T_{in} = 60$ °C |
| Fluid volumetric heat capacity[2] | $C_{pf} = 4.2 \cdot 10^6$ J/(m³ °C) |
| Rock volumetric heat capacity[2] | $C_{pb} = 3.12 \cdot 10^6$ J/(m³ °C) |
| Rock thermal conductivity[2] | $K_b = 3$ W/(m °C) |
| Calcite rate coefficient[3] | $\lambda = 10^{-6}$ m/s |
| Silica rate coefficient[4] | $\lambda = 5 \cdot 10^{-10}$ m/s |
| Fractured carbonates specific reactive surface area[5] | $A_s = 10$ m⁻¹ |
| Porous sandstones specific reactive surface area[6] | $A_s = 10^4$ m⁻¹ |
| Calcite mineral concentration[3] | $c_{sol} = 2.7 \cdot 10^4$ mol/m³ |
| Silica mineral concentration[4] | $c_{sol} = 3.7 \cdot 10^4$ mol/m³ |
| Solubility change parameter calcite[7] | $\beta = -0.075$ mol/(m³ °C) |
| Solubility change parameter silica[1] | $\beta = 0.04$ mol/(m³ °C) |
| Stoichiometry coefficient[3,4] | $v = 1$ |
| Exponent of $\theta$-$k$ relation[5] | $n = 2$-$20$ |

1-(Glassley, 2014); 2-(Huenges and Ledru, 2011); 3-(Palmer, 1991); 4-(Rimstidt and Barnes, 1980); 5- see text; 6-(Lai et al.,
2015); 7-(Roded et al., 2023).

In the simulations, we assign characteristic porosity ($\theta$), and reactive surface area, ($A_s$,) for the different aquifer types. In accordance with common field observations, we consider a carbonate aquifer in which flow and dissolution are focused in the permeable fracture network, and a porous sandstone aquifer characterized by high intergranular permeability (Jamtveit and Yardley, 1996; Bear and Cheng, 2010).
The different aquifer characteristics are reflected in significant differences in $\theta$ and $A_s$ for the different aquifer types. Specifically, carbonates are often characterized by permeability contrasts spanning orders of magnitudes between the fractures and the rock matrix (Dreybrodt et al., 2005; Lucia, 2007). Consequently, transport in the matrix occurs mostly by slow diffusion and the reaction within the matrix can be neglected. Hence, solely the reactive surface area, $A_s$, of the fractures effectively participates in
the reaction (Deng and Spycher, 2019; Maher et al., 2006; Seigneur et al., 2019; Pacheco and Alencoão,





2006). In this case, the $\theta$ can be minimal (Lucia, 2007) and $A_s$ is orders of magnitude smaller compared to its value in porous sandstones (Lai et al., 2015; Pacheco and Alencoão, 2006; Pacheco and Van der Weijden, 2014; Seigneur et al., 2019). This disparity can lead to substantial differences in characteristic alteration rates and Damköhler numbers in these systems (Ladd & Szymczak, 2021; Lucia, 2007;

Seigneur et al., 2019). Here, it is further assumed that the fracture density is high, and the network is of high connectivity allowing it to be treated as a continuum (Anderson et al., 2015; Sahimi, 2011).

We consider here an injection flow rate of $Q = 500$ m$^3$/day, which falls within the typical range of flow rates observed in relevant applications, such as geothermal systems (Glassley, 2014) or groundwater storage and recovery (Maliva, 2019). The injection temperature is set to $T_{in} = 60\,°C$ and aquifer ambient

temperature is set to $T_0 = 20\,°C$ ($\Delta T = 40\,°C$).

**3.2 Carbonate Aquifer Dissolution by Cooling Water**

In Fig. 2, the results of CO$_2$-rich hot water injection into a carbonate aquifer at successive times since the beginning of the injection are shown (Eqs. 10, 15, D.2 and 17 are solved for $t = 0.2$, 10 and 100 kyr). During the radial flow within the aquifer, the hot fluid cools by transferring heat into the confining

layers, which heat up with time, resulting in the gradual advancement of the thermal front downstream (Fig. 2a). The cooling induces solute disequilibrium ($\Lambda$) associated with undersaturation (note that $\Lambda$ is negative for undersaturation and positive for supersaturation, see Eq. 6). The magnitude of $\Lambda$ in the aquifer is small compared to the absolute solubility change in the system, $\Delta c_s = |c_s(T_{in}) - c_s(T_0)|$, i.e., between $c_s(T_{in})$ at the injection point to $c_s(T_0)$ at ambient conditions ($|\Lambda|/\Delta c_s << 1\%$, see Fig. 2b). The

small magnitude of disequilibrium is associated with relatively high PCO$_2$ considered here (0.03 MPa) and rapid kinetics under these conditions. The quasi-equilibrium conditions may allow simplification and calculation of the local reaction rate from transport processes alone, regardless of kinetics, referred to as the so-called *equilibrium model* (Andre and Rajaram, 2005; Bekri et al., 1995; Golfier et al., 2002; Lichtner, 1991), which will be the subject of a future publication.




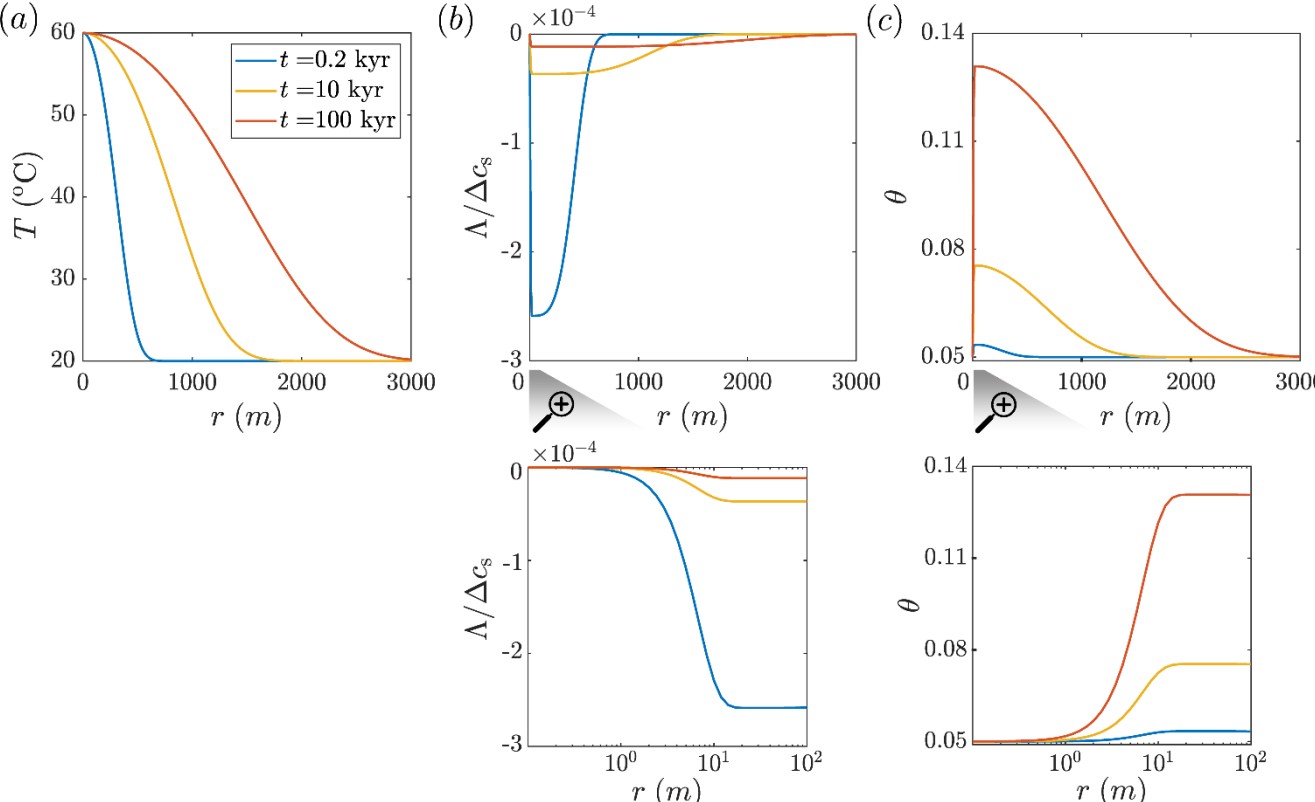

**Figure 2: Carbonate aquifer dissolution by cooling hot water.** Temperature, $T$, solute disequilibrium, $\Lambda$, and porosity, $\theta$ in the aquifer are plotted as functions of radial position, $r$, at different times (computed using Eqs. 10, 15, D.2 and 17). (a) The hot flow cools gradually as it travels through the aquifer, transferring heat to the confining rocks, thereby causing them to warm over time and the thermal front to progress downstream. (b) Cooling induces undersaturation (negative disequilibrium, $\Lambda$, see Eq. 6), which is of a relatively small magnitude due to the rapid kinetics of calcite dissolution. $\Lambda$ is normalized by the total solubility change in the system, $\Delta c_s$, (refer to the text for $\Delta c_s$ definition). The water is hot and saturated at the inlet, $c = c_s(T_{in})$. Undersaturation quickly develops near the inlet ($r \approx 20$ m, as shown in the magnification) and then gradually diminishes due to the dissolution reactions further along the flow path ($\Lambda$ approaches zero). As the thermal front propagates over time, and thermal gradients diminish, the $\Lambda$ curves also flattens. (c) Corresponding to $\Lambda$ variations, a porosity profile develops over time (see the magnification for the inlet-adjacent region).

Although the magnitude of disequilibrium, $\Lambda$, is small, it controls the alteration of the aquifer and the evolution of its properties. Significantly, because the water at the inlet is hot and saturated with calcite,





$c = c_s(T_{in})$, disequilibrium and the reaction rate are zero at the inlet leading to no change in the porosity (see Fig. 2b and 3c and their magnifications). Disequilibrium (undersaturation) sharply develops downstream from the injection site forming first a small minimum (at $r \approx 20$ m) and gradually increasing to zero at greater distances. Undersaturation and dissolution along the flow path are

controlled by the interplay of three processes: (I) dissolution reducing undersaturation (i.e., $\Lambda$ becomes closer to zero), (II) progressive cooling increasing undersaturation, and (III) advection transporting reaction products (i.e., calcium ions) radially outward from the well, helping maintain undersaturation. Here, the effect of fluid velocity and advection decays with a distance as $1/r$.

High advection and cooling rates near the inlet result in the abrupt formation of undersaturation (i.e.,

negative $\Lambda$). Further downstream, undersaturation diminishes due to dissolution reactions. As the thermal front advances downstream over time and the temperature gradients diminish along the aquifer, the $\Lambda$ curve flattens and becomes more elongated (see curves for $t = 10$ and $100$ kyr in Fig. 2b). Due to the disequilibrium, porosity grows with time. The porosity profile sharply increases near the inlet and then gradually decreases downstream (Fig. 2c). The porosity changes are extensive and take place over

an aquifer area of $\sim 30$ km$^2$ within a relatively short geological timescale of $100$ kyr, resulting in the addition of significant void space of thousands of cubic meters ($\sim 5 \cdot 10^3$ m$^3$).

An essential assumption underlying the solutions in section 2 and the results depicted in Fig. 2, is the assumption of spatial uniformity and symmetry of reactive flow. In practical scenarios, however, dissolutional instabilities at the reaction front can emerge. These instabilities, owing to the positive

feedback between reaction and transport, may evolve into dissolution channels, often referred to as *wormholes* (Aharonov et al., 1997; Budek & Szymczak, 2012; Chadam et al., 1986; Ortoleva et al., 1987b; Roded et al., 2021). The wormholes concentrate reactive flow, resulting in heterogeneous flow fields that cannot be accurately represented by assuming symmetry and uniformity. In such a case, the results of Fig. 2 can only be regarded as an average solution, which is not accurate locally.

Isothermal dissolution, driven by undersaturation of the incoming solution is known to be unstable in the radial geometry for large enough solute Péclet, $Pe_s$, numbers and intermediate Damköhler numbers.





The Damköhler number here is given by $Da = A_s \lambda l_A / u$, and represents the ratio between advective and reactive timescales (Daccord, 1987; Kalia and Balakotaiah, 2007; Grodzki and Szymczak, 2019; Xu et al., 2020). However, in our case, cooling of the solution leads to its renewed aggressiveness, hence extending the penetration length in the system which may influence the stability of the reactive front (Xu et al., 2020). The effect of renewed aggressiveness by considering solubility gradients was studied for planer reactive flow in Aharonov et al. (1997) and Spiegelman et al. (2001), but requires further investigation for radial flow, and taking into account coupling with heat transfer.

**3.3 Silica Precipitation by Cooling Water**

Here, we consider the injection of hot silica-rich water that cools, becoming supersaturated and leading to silica precipitation, consequently reducing void-space and permeability. While the previous case involved dissolution, this one involves precipitation; however, the thermal and reactive transport processes are similar in both cases (with approximately mirror image $\Lambda$ and $\theta$ profiles, c.f., Fig.2b-c and Fig.3a-b).

Similar to the previous section, the low magnitude of $\Lambda$ suggests that the reaction rate (Eq. 5) is relatively high compared to transport processes, effectively reducing disequilibrium, $\Lambda$. It is noted that the reaction rates are high in both systems, despite the orders of magnitude differences in the kinetic rate coefficient ($\lambda = 10^{-6}$ m/s for calcite dissolution compared to $5 \cdot 10^{-10}$ m/s for silica precipitation). However, this difference is largely compensated by the contrast between the reactive surface area of the porous sandstone and fractured carbonate aquifers ($A_s = 10^4$ m$^{-1}$ compared to 10 m$^{-1}$, respectively). It should also be noted that while precipitation of crystalline and non-crystalline silica (amorphous) is characterized by relatively high rates, dissolution of quartz and silica polymorphs is typically slower by several orders of magnitudes (Rimstidt and Barnes, 1980).

While the reaction rates are high in both systems, differences exist in the absolute amount of porosity change resulting from the injection. For example, the maximal porosity change in the aquifer due to silica precipitation is approximately $\Delta\theta_{max} \approx 0.03$, whereas for the carbonate case it is around $\Delta\theta_{max} \approx 0.08$ (where $\Delta\theta_{max} = |\theta_{max}(t = 100 \text{ kyr}) - \theta_0|$, and $\theta_{max}$ denote the maximal porosity change along the





profile). The predicted lower porosity change in silica arises mostly due to its lower total solubility

change, $\Delta c_s$, and the reduced dependence of mineral solubility on temperature, expressed here by the $\beta$

parameter (see Table 2). This conclusion is further supported by the fact that no disequilibrated fluid

exits the system: the fluid outflows from the system at $r = 3000$ m, at a temperature close to the ambient

temperature, $T_0$, (Fig. 2a) and chemically equilibrated ($\Lambda = 0$; Fig. 2b and Fig. 3a).

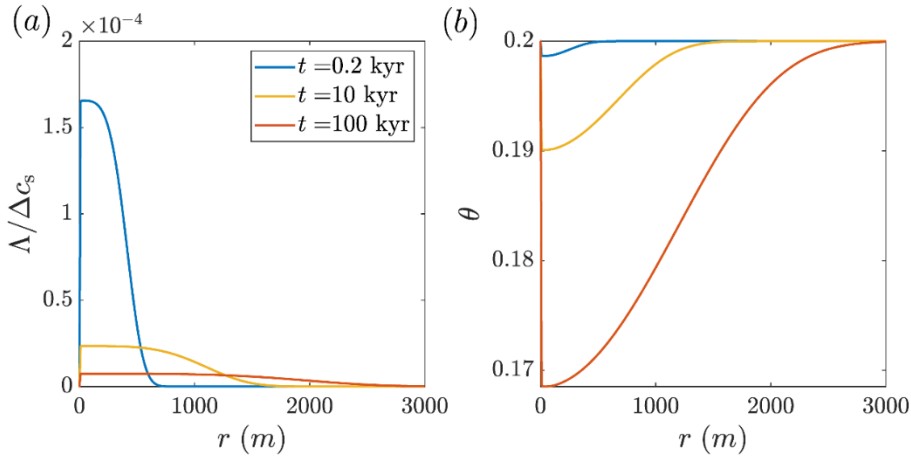

**Figure 3: Silica precipitation in sandstone aquifer by cooling hot water.** The calculated solute disequilibrium,

$\Lambda$, and porosity, $\theta$, as functions of the lateral position, $r$, are shown at different times since the beginning of the

injection (calculated using Eqs. 15, D.2 and 17; the temperature profile is given in Fig. 2a). The reactive transport

processes in this case are similar to the carbonate dissolution system shown in Fig. 2, with insets Fig. 2b-c being

approximately mirror images of (a) and (b), showing supersaturation and porosity reduction. (a) As a result of

cooling, solute disequilibrium corresponding to supersaturation ($\Lambda$, Eq. 6) develops, which is of small magnitude

due to the high reaction rates ($\Lambda$ is scaled by the total solubility change in the system, $\Delta c_s$, refer to the text for $\Delta c_s$

definition). The water enters hot and saturated at the inlet, $c = c_s(T_{in})$, and, subsequently, $\Lambda$ increases rapidly and

then gradually diminishes downstream due of the reaction. The advancement of the thermal front over time and

lower gradients lead to the flattening of $\Lambda$ curves. (b) In accordance with $\Lambda$, an extensive porosity profile develops

over time.

## 3.4 Permeability Evolution of the Aquifers


The porosity changes affect the aquifer hydraulics. Here, we calculate the effective aquifer

permeability, $k_{eff}$, within a distance, $R$, around the well. $k_{eff}$ is calculated based on the relationship




between the local porosity and permeability, utilizing the power-law relation $k(r)/k_0 = (\theta(r)/\theta_0)^n$, where $k_0$ and $\theta_0$ are the initial permeability and porosity (the steps for the calculation of $k_{\text{eff}}$ are presented in

Appendix E). The exponent $n$ depends on various factors such as medium microstructural details and the nature of the alteration processes (Hommel et al., 2018; Seigneur et al., 2019; Steefel et al., 2015). The limited predictive capabilities of $k$-$\theta$ relations was previously noted (e.g., Sabo & Beckingham, 2021). Here it is applied to merely evaluate the general trends.

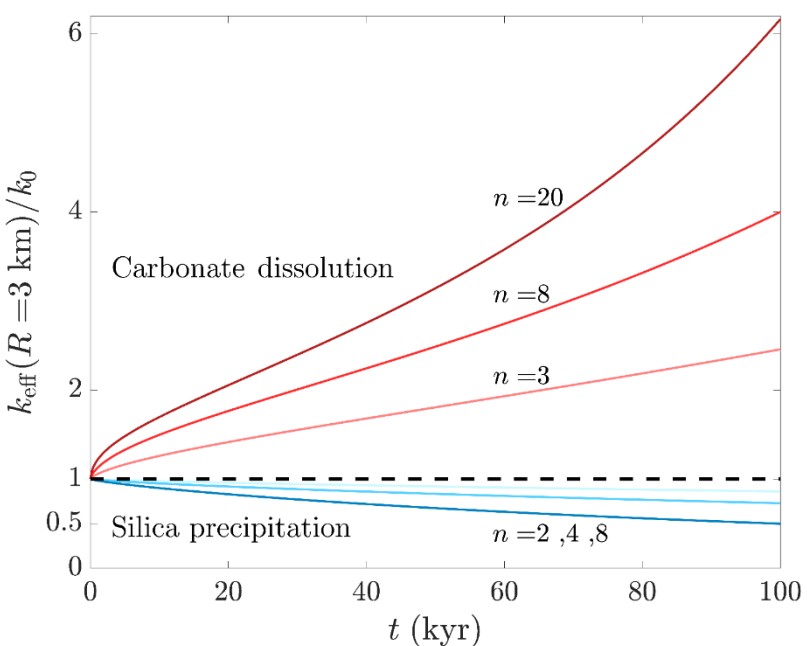

**Figure 4: Evolution of aquifer effective permeability due to dissolution and precipitation.** The effective permeability is $k_{\text{eff}}$, and $t$ is time; red and blue curves designate carbonate dissolution and silica precipitation, respectively. $k_{\text{eff}}$ is calculated within radius $R = 3$ km from the well and is normalized by its initial value, $k_0$. The power-law $\theta$-$k$ relation is used to determine $k_{\text{eff}}$ from the local porosity, $\theta(r)$, and permeability, $k(r)$, with typical exponent values of $n = 3$-20 for dissolution, and $n = 2$-8 for precipitation. $k_{\text{eff}}$ can be substantially altered in

carbonate aquifers due to dissolution even within tens to hundreds of years, while tens of thousands of years are required for similar magnitudes of change by silica precipitation.

The wide range of heterogeneous microstructures in rocks and sediments, and their response to different reactive flow regimes, leads to a large variability in the exponent $n$ values. For example, for relatively





uniform spatial dissolution, $n$ can range from ~3 to a few dozen for the early stages of flow or when
wormholes develop (Hao et al., 2013; Noiriel et al., 2005; Roded et al., 2020). For precipitation, $n$
typically ranges from ~2 and up to above 10 (Aharonov et al., 1998; Hommel et al., 2018; Seigneur et
al., 2019).

Figure 4 shows $k_{eff}$ evolution over time for representative exponent values within a distance, $R = 3$ km.
The rapid increase in carbonate aquifer permeability indicates (in agreement with previous works; Agar
& Geiger, 2015; Andre & Rajaram, 2005; Dreybrodt et al., 2005) that $k_{eff}$ can be substantially altered
within relatively short geological timescales. Specifically, the results suggest that $k_{eff}$ can even increase
by several tens of percents within tens to hundreds of years. Conversely, significant $k_{eff}$ alterations due
to silica precipitations (10-50 % reduction) involve typical timescales of alterations by fluid-rock
interaction of tens of thousands of years. These findings are consistent with previous observations of
dissolution and precipitation under solubility gradient (e.g., Aharonov et al., 1997), emphasizing
differences between these processes, as embodied in the exponent $n$.

## 4 Summary and Conclusions

In this paper, we considered non-isothermal injection into a confined aquifer, and the settings and
solution of the so-called Lauwerier problem, to derive coupled thermally-driven reactive transport
solutions (*reactive Lauwerier problem*). The presented solution is one of very few existing analytical
solutions in the field of thermo-hydro-chemical flows in porous media. The thermo-hydro-chemical
scenarios considered here involved geochemical disequilibrium and reactions induced by thermally-
driven solubility changes, leading to mineral dissolution or precipitation. In the first section, solutions
were derived for the evolution of solute concentration in radial and planar cases. These derivations
utilized the non-isothermal Lauwerier solution to calculate the temperature-dependent solubility, which
was then substituted into the reactive transport equation. Subsequently, the obtained concentration
profiles were used to derive expressions for the porosity change in the aquifer.

In the second section, these solutions were employed to study two common cases in geothermal and
water resource systems, exhibiting opposite feedbacks on porosity evolution: (I) injection of hot $CO_2$-





rich water into a fractured carbonate aquifer, leading to cooling and dissolution, and (II) injection of hot silica-rich water into sandstone aquifer leading to silica precipitation. The resulting porosity profiles were then used to calculate the hydraulic changes and effective aquifer permeabilities. The results show that the timescale of porosity development in these systems is of the order of thousands to dozens of thousands of years, depending on the THC conditions (in agreement with previous works; Andre and

Rajaram, 2005; Roded et al., 2023). Despite the often-faster kinetics of carbonate dissolution compared to silica precipitation, similar timescales are observed in both systems. This is attributed to the high specific reactive surface area of sandstones, which enhances the reaction rate, compensating for the differences in kinetics between carbonate dissolution and silica precipitation. However, substantial hydraulic changes occur much faster in carbonate aquifers, possibly within tens to hundreds of years,

primarily due to the rapid enhancement of permeability resulting from dissolution.

It is worth noting that under the typical conditions considered, the reaction rates are high and the geochemical disequilibrium in these systems is minimal (i.e., quasi-equilibrium). In such conditions, the *equilibrium assumption* can be applied which simplifies calculations in reactive Lauwerier problem and comprises an ongoing area of inquiry. The solutions and analyses provided contribute to the

understanding of natural and engineered hydrothermal systems, such as aquifer storage and recovery (ASR) and thermal energy storage (ATES) applications. Additionally, these solutions can aid in the development and benchmarking of coupled numerical models.






## Appendices

### Appendix A: An Extended Form of the Conservation Equations

**Aquifer temperature.** Heat transport through the rocks confining the aquifer is governed by conduction. Assuming radial symmetry, the heat equation in polar coordinates becomes

$$\frac{\partial T}{\partial t} = \frac{\alpha_b}{r}\frac{\partial}{\partial r}\left(r\frac{\partial T}{\partial r}\right) + \alpha_b\frac{\partial^2 T}{\partial z^2}, \qquad \begin{cases} z \leq -\dfrac{H}{2} \\[2mm] z \geq \dfrac{H}{2} \end{cases}, \tag{A.1}$$


where $T$ is the temperature, $t$ is time, $r$ and $z$ are the radial and vertical coordinates, respectively, with their origin at the injection well center, and $H$ is aquifer thickness (see Fig. 1). The quantity $\alpha_b = K_b/(C_{p_b})$ is the thermal diffusivity, where the subscript b denotes rock, $K$ is the thermal conductivity, and $C_p$ is the volumetric heat capacity (Stauffer et al., 2014).

Assuming that heat transport in the fluid within the aquifer is governed by advection and conduction, the heat-transport equation can then be expressed as

$$C_{p_b}\frac{\partial T}{\partial t} = -C_{p_f}\frac{1}{r}\frac{\partial(ruT)}{\partial r} + K_b\left(\frac{1}{r}\frac{\partial}{\partial r}\left(r\frac{\partial T}{\partial r}\right) + \frac{\partial^2 T}{\partial z^2}\right), \quad \text{for} \quad -\frac{H}{2} \leq z \leq \frac{H}{2}, \tag{A.2}$$

where subscript f denotes fluid, $u(r)$ is the fluid velocity (or Darcy flux) and can be calculated from the total volumetric flow rate $Q$ using $u = Q/(H2\pi r)$ (assuming uniformity of $u$ along the $z$ direction of the

aquifer; Andre & Rajaram, 2005; Chaudhuri et al., 2013).

Assuming complete thermal mixing in the transverse direction ($z$) of the aquifer, allows to establish the "depth-averaged" Eq. 2 in the main text. In this case, the heat exchange between the aquifer and the confining rocks is integrated within the heat exchange term ($\Theta$).

**Reactive Transport.** Similarly, the solute transport advection-diffusion-reaction equation in the aquifer

is

$$\frac{\partial c}{\partial t} = -u\frac{\partial c}{\partial r} + D\left(\frac{1}{r}\frac{\partial}{\partial r}\left(r\frac{\partial c}{\partial r}\right) + \frac{\partial^2 c}{\partial z^2}\right) - \Omega(r,t), \quad \text{for} \quad -\frac{H}{2} \leq z \leq \frac{H}{2}, \tag{A.3}$$





where $c$ is the solute concentration [M/L$^3$], $D$ is the molecular diffusion coefficient, and $\Omega$ is the reaction term (Chaudhuri et al., 2013; Szymczak and Ladd, 2012). The equations describing the reaction term, $\Omega$, saturation concentration, $c_s$, dependence on the temperature and the porosity change are given in section 2.3 in the main text (Eqs. 5, 7 and 8, respectively).

In the case of planar flow and Cartesian coordinates the equations A.1-A.3 above take the form,

$$\frac{\partial T}{\partial t} = \alpha_b \left( \frac{\partial^2 T}{\partial x^2} + \frac{\partial^2 T}{\partial z^2} \right), \qquad \begin{cases} z \leq -\dfrac{H}{2}, \\ z \geq \dfrac{H}{2} \end{cases} \tag{A.4}$$

$$C_{p_b} \frac{\partial T}{\partial t} = -u C_{p_f} \frac{\partial T}{\partial x} + K_b \left( \frac{\partial^2 T}{\partial x^2} + \frac{\partial^2 T}{\partial z^2} \right), \quad \text{for} \quad -\frac{H}{2} \leq z \leq \frac{H}{2}, \tag{A.5}$$

and

$$\frac{\partial c}{\partial t} = -u \frac{\partial c}{\partial x} + D \left( \frac{\partial^2 c}{\partial x^2} + \frac{\partial^2 c}{\partial z^2} \right) - \Omega(x, t), \quad \text{for} \quad -\frac{H}{2} \leq z \leq \frac{H}{2}. \tag{A.6}$$

**Appendix B: Timescales Analysis to Validate the Quasi-static Assumption**

In our reactive transport calculations and Eq. 4 used for developing the solutions in section 2, we adopt the quasi-static approach (Detwiler & Rajaram, 2007; Ladd & Szymczak, 2017; Lichtner, 1991; Roded, Aharonov, Holtzman, et al., 2020; Sanford & Konikow, 1989) and neglect the transient term (present in Eq. A.3).. However, it is noted that temporal variations do take place due to changes in the temperature field and its effect on the solubility, which are accounted for by coupling the equations.

The justification for the quasi-static assumption lies in the significant separation of characteristic timescales in the system. There are three important timescales in our problem: (I) the timescale governing reactant transport ($t_A = l_A/u$), (II) mineral chemical alteration time ($t_M$), and (III) the characteristic conduction heat transport time ($t_C = l_C^2/\alpha_b$). The latter affects the solubility of aquifer minerals, thus influencing reaction and solute transport. Specifically, the conditions for the validity of quasi-static assumption are that $t_C$ and $t_M$ are several orders of magnitude larger compared to reactant





transport relaxation time, $t_A$ (i.e., $t_A << t_M$ and $t_A << t_C$). For instance, in relatively fast-reacting natural carbonate systems the doubling of initial pore size or fracture aperture typically occurs over a timescale of months to years. In silicate minerals, these timescales are of the order of thousands of years (Dove & Crerar, 1990; Ladd & Szymczak, 2021; Szymczak & Ladd, 2012; Zhu, 2005). Similarly, the timescale characteristics for the conduction processes in the confining rocks ($t_C$) are commonly several orders of

magnitude longer than the relaxation times for reactant transport ($t_A$), which essentially maintains a steady-state throughout the aquifer evolution.

The timescale of mineral alteration is given by $t_M = \delta\theta/\gamma A_s\lambda$, where $\delta\theta$ represents a minute change in porosity, $A_s$ stands for the specific surface area of the reacting mineral [$L^2/L^3$] and $\lambda$ is the kinetic reaction rate coefficient [L/T]. Here, $\gamma = \Delta c_s/c_{sol}v$, where $c_{sol}$ is the mineral concentration in the solid, $v$

accounts for the stoichiometry of the reaction and $\Delta c_s$ is the variation in solubility induced by thermal changes along the flow path. $\Delta c_s$ is calculated here from the difference between the injected saturated fluid concentration, $c(\varphi=0) = c_s(T_{in})$, and the downstream saturation at the background aquifer temperature, $c = c_s(T_0)$ (i.e., $\Delta c_s = |c_s(T_{in})-c_s(T_0)|$). $\gamma$ is often referred to as the acid capacity number, representing the ratio between (I) the maximum number of molecules in a unit volume of fluid

dissolving or precipitating mineral from the fluid along the flow path (calculated from the ratio, $\Delta c_s/v$), (II) to the number of molecules in a unit volume of a mineral, $c_{sol}$ (see parameter values in Table 2; Ladd & Szymczak, 2017; Roded, Aharonov, Holtzman, et al., 2020).

In the calculation of the timescale $t_A$, the characteristic length scale, $l_A$, can be set equal to the reactive front length, which in turn is affected by the thermal front length along the aquifer ($\varphi$-direction). The

length scale $l_C$ (used in $t_C$ calculation) corresponds to the thermal front that develops in the confining insulating layers in the $z$-direction, which elongates over time. In practice, the timescale separation between $t_A$ and $t_M$ and $t_C$, can also be validated *a posteriori*. Under a large set of conditions, the reaction rate is limited solely by advective transport (i.e., regardless of kinetics), which leads to small geochemical disequilibrium (Andre & Rajaram, 2005). In such conditions, the actual timescale of

matrix deformation will be much longer than predicted by the expression given above for $t_M$.



In this appendix, the solution of Eq. 10 is compared to its approximated solution, when $t' \approx t$ is assumed (Fig. A1). The results demonstrate that for times longer than 100 years, the differences between the solutions diminish, with a maximal error of 1.5 %, where the error is defined as $Err = 100*(|T_{Ext} - T_{Apr}|)/\Delta T$, with $T_{Ext}$ and $T_{Apr}$ being the exact and approximated solutions. These results confirm the validity of the assumption of $t' \approx t$ and the derived solutions for times longer than 100 years under the conditions considered.

## Appendix C: Lauwerier Solution Validity Assuming $t' \approx t$

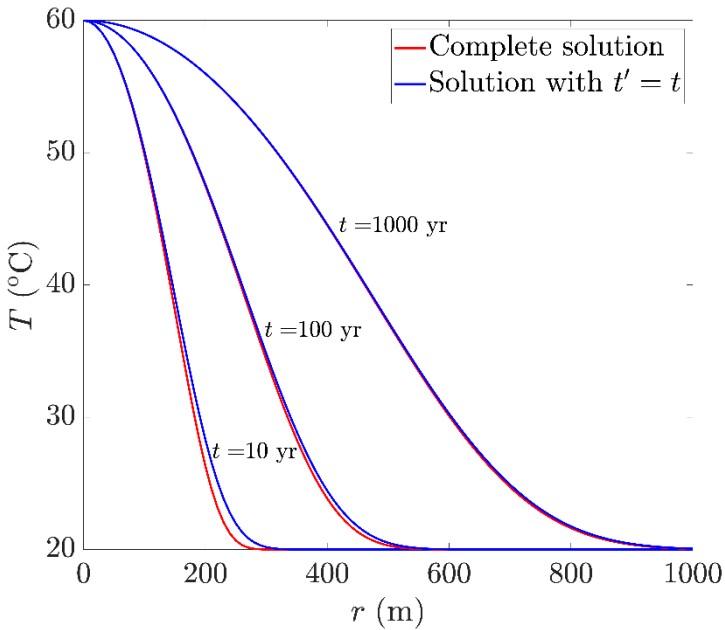

**Figure A1: Comparison of the full and approximate solution for the temperature profile.** The approximate solution considers $t' = t$ (Eq. 10). The results demonstrate that for times longer than 100 years, the differences between the solutions diminish, with a maximal error of 1.5% (see text).

## Appendix D: Asymptotic Expansion for the Disequilibrium Solutions

To obtain a solution by computational means and prevent an integer overflow (Press et al., 2007), it is useful to derive an approximate solution for Eq. 15 using the first-order asymptotic expansion of erfc. Substituting this expansion into Eq. 15 leads to




$$\Lambda = \frac{\Delta T\beta}{\sqrt{\pi}} e^{\left(\frac{\eta^2}{4\zeta^2}-\eta r^2\right)}\left(-e^{\left(-\frac{\eta^2}{4\zeta^2}\right)}\frac{2\zeta}{\eta} + e^{-\left(\frac{\eta^2}{4\zeta^2}-\eta r^2+\zeta^2 r^4\right)}\frac{1}{\frac{\eta}{2\zeta}-\zeta r^2}\right),\qquad (D.1)$$

and after further rearrangement, we finally arrive at:

$$\Lambda = \frac{\Delta T\beta}{\sqrt{\pi}} e^{(-\eta r^2)}\left(\frac{e^{(\eta r^2-\zeta^2 r^4)}}{\frac{\eta}{2\zeta}-\zeta r^2} - \frac{2\zeta}{\eta}\right).\qquad (D.2)$$

For the planar injection case, we obtain from Eq. 20,

$$\Lambda = \frac{\Delta T\beta}{\sqrt{\pi}} e^{(-\sigma x)}\left(\frac{e^{(\sigma x-\omega^2 x^2)}}{\frac{\sigma}{2\omega}-\omega x} - \frac{2\omega}{\sigma}\right).\qquad (D.3)$$

Eq. D.3 is used to obtain a solution for longer times presented in section 3.

## Appendix E: Permeability of an Aquifer with Nonuniform Porosity Profile

Using Darcy's law, we calculate an effective permeability, $k_{\text{eff}}$, for the aquifer around the well within a radius $r = R$. The Darcy's law under these conditions is

$$u(r) = -\frac{k(r)}{\mu}\frac{dp}{dr},\qquad (E.1)$$

where $p$ and $\mu$ are the fluid pressure and viscosity and $k$ permeability. Integrating Eq. E.1 between $r=0$ and $r=R$ leads to

$$u(R) = -\frac{R}{\mu\int_0^R \frac{dr}{k(r)}}\left(\frac{\Delta p}{R}\right),\qquad (E.2)$$

and the effective permeability is



$$k_{\text{eff}} = \frac{R}{\int_0^R \frac{\mathrm{d}r}{k(r)}},$$
$(E.3)$

which is calculated by numerical integration over the porosity profile and the power-law given in section 3.4.

**Code & Data availability:**

Codes and data produced in this study are available from the corresponding authors upon request (will
be made available online prior to publication)

**Author contribution:**

Theoretical analysis: E.A., R.R., P.S. Conceptualization: R.R., E.A., P.S. Numerical analysis: R.R. Geochemical modelling: R.R., B.L., L.E.D., Writing—original draft: R.R. Writing—review & editing: R.R., E.A., L.E.D., P.S., M.V., B.L.

**Competing interests:**

Authors declare that they have no competing interests.

**Acknowledgments:**

ISF grant # 910/17 (E.A.); National Science Centre (NCN; Poland) under CEUS-UNISONO grant 2020/02/Y/ST3/00121 (P.S).




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
