# Peer review of "Solutions for Thermally-driven Reactive Transport and Porosity Evolution in Geothermal Systems ("Reactive Lauwerier Problem")"

_Hydrology and Earth System Sciences, 2023_

## Author Comment (AC1)

**Referee #1**

We thank the Reviewer for the careful review and comments, which will help us improve the manuscript. We are glad the Reviewer found the manuscript to be well-written, well-organized, and noted the topic is of high interest to the scientific community. In what follows, we respond in turn to each of the comments (included verbatim, in **bold**), attached also as a separate file.

1. **I think the Introduction is a bit wordy and can be summarized in 4-5 concise paragraphs raising the main points.**

Following the Referee's comment, the introduction will be shortened while retaining important background information and necessary details for the presentation.

2. **Considering the main model assumptions, how applicable/reliable is constant fluid density with temperature changes, in particular, for CO2 as the working fluid? The same question will be raised for the assumption of having the same heat capacity for both confining rocks and the aquifer.**

We thank the Reviewer for noting these issues. The Lauwerier solution (Lauwerier, 1955) refers to a thin confined aquifer layer, in which no substantial vertical temperature gradients develop (lines 211-213). Consequently, under these conditions, no free convection or convection cells are expected to develop.

Regarding the potential effect of variable density on lateral flow, changes in density due to heating or cooling along the flow path can cause variations in flow rate. Specifically, fluid contraction during cooling can lead to a decrease in flow rate, while fluid expansion during heating can lead to an increase in flow rate along the path. However, in most cases, the overall temperature change does not exceed $\Delta T = 80$ °C, resulting in a low change in water density, typically around 2%. Consequently, this change does not have a substantial effect on flow and reactive transport (assuming there is no phase change). However, for the case of supercritical $CO_2$, the changes in density can be much larger, and in some cases the incompressibility assumption may not be appropriate.

Regarding the uniform heat capacities, the reactive Lauwerier solution, like the original Lauwerier solution, does not inherently assume uniform heat capacities. This assumption is made here for simplicity and to avoid cumbersome equations. By adopting alternative definitions of the parameters in Eqs. 10 and 19, non-uniform heat capacities of the confining layers and the aquifer can be considered in reactive Lauwerier solution (see Eq. 3.122 in Stauffer et al. (2014).

The above explanations, specifically noting the limitation of the assumption with respect to scenarios where $CO_2$ is the working fluid, will be integrated in the revised manuscript.

3.  **The authors have used a power-law relationship between the porosity and permeability of the aquifer to calculate the effective permeability. The authors have raised the limited predictive capability of these types of relations as well and mentioned that they solely used this type of relations to evaluate the general trends. Although this type of relationship could help evaluate the general trend of permeability evolution in mineral dissolution cases, they may not be the best choice for anti-correlated porosity-permeability changes in the systems when the percolating fluid has a weak capacity for dissolution or when we have mineral precipitation. Why did the authors not use other types of porosity-permeability relationships (e.g., a two-parameter exponential model) for sandstone aquifers where they expected to see mineral precipitation?**

We agree with the Reviewer that for specific cases and reactive flow duration (e.g., Garing et al., 2015) there may be more appropriate choices for porosity-permeability relations. However, we note this falls outside the scope of the current work. Here, we emphasize the different trends of aquifer permeability evolution due to dissolution in a carbonate aquifer and silica precipitation. Consequently, we selected a generic form of the porosity-permeability relation. The power-law relation demonstrates that permeability will typically rise within relatively short geological timescales in carbonate aquifers due to dissolution while substantial permeability changes due to silica precipitation occur on the order of tens of thousands of years. We argue, besides unique cases, the conclusions will remain valid regardless of the porosity-permeability relation used.

4.  **It seems that the last sentence in the Figure 1 caption is not complete. Please review the caption and correct it.**

Thank you for catching this oversight, the caption for Figure 1 will be corrected.

5.  **How reliable are the utilized values for the surface area of the carbonates and sandstones? Are they within the reported range for the surface area of these rocks in the literature?**

The values for the specific surface area of the sandstones utilized in the present work (i.e., $A_s = 10^4 \, m^{-1}$) fall within the ranges reported in the literature for common sandstone formations ($10^4$-$10^5 \, m^{-1}$; Hussaini & Dvorkin, 2021; Lai et al., 2015). However, it is important to note the ongoing research and discussions regarding the uncertainties in the estimation of reactive surface area in various rocks and under different reactive transport conditions (Noiriel & Daval, 2017; Recalcati et al., 2024; Seigneur et al., 2019). To address the Reviewer's comment, the range of values reported in the literature including additional references will be added in the revised manuscript.

For fractured carbonates, previous studies have demonstrated that in cases of large permeability contrast between fractures and the rock matrix, only the fracture surface area effectively participates in the reaction (i.e., constitutes the reactive surface area). This has been demonstrated by field case studies (e.g., MacQuarrie & Mayer, 2005; Pacheco & Alencoão, 2006). In this case, the reactive surface area can be calculated using $A_s = 2 \cdot \kappa \cdot RF$ where $\kappa$ is fracture density (defined as the number of fractures per unit volume), the factor of 2 accounts for the presence of two surfaces, and RF is the roughness factor (see Deng et al., 2018). Assuming $\kappa = 1/3^3 \, m^{-3}$ and RF = 1.35, results in $A_s = 0.1 \, m^{-1}$. Typical values of $\kappa$ and fracture spacing can span a substantial range and may be higher or lower (Narr & Suppe, 1991; Scholz, 2019). Following the Reviewer's comment, this issue will be further clarified and references will be added to the revised manuscript.

**References**

Deng, H., Molins, S., Trebotich, D., Steefel, C., & DePaolo, D. (2018). Pore-scale numerical investigation of the impacts of surface roughness: Upscaling of reaction rates in rough fractures. *Geochimica et Cosmochimica Acta*, *239*, 374–389.

Garing, C., Gouze, P., Kassab, M., Riva, M., & Guadagnini, A. (2015). Anti-correlated porosity–permeability changes during the dissolution of carbonate rocks: Experimental evidences and modeling. *Transport in Porous Media*, *107*(2), 595–621.

Hussaini, S. R., & Dvorkin, J. (2021). Specific surface area versus porosity from digital images. *Journal of Petroleum Science and Engineering*, *196*, 107773.

Lai, P., Moulton, K., & Krevor, S. (2015). Pore-scale heterogeneity in the mineral distribution and reactive surface area of porous rocks. *Chemical Geology*, *411*, 260–273.

Lauwerier, H. (1955). The transport of heat in an oil layer caused by the injection of hot fluid. *Applied Scientific Research, Section A*, *5*(2), 145–150.

MacQuarrie, K. T., & Mayer, K. U. (2005). Reactive transport modeling in fractured rock: A state-of-the-science review. *Earth-Science Reviews*, *72*(3–4), 189–227.

Narr, W., & Suppe, J. (1991). Joint spacing in sedimentary rocks. *Journal of Structural Geology*, *13*(9), 1037–1048.

Noiriel, C., & Daval, D. (2017). Pore-scale geochemical reactivity associated with CO2 storage: New frontiers at the fluid–solid interface. *Accounts of Chemical Research*, *50*(4), 759–768.

Pacheco, F. A. L., & Alencoão, A. M. P. (2006). Role of fractures in weathering of solid rocks: Narrowing the gap between laboratory and field weathering rates. *Journal of Hydrology*, *316*(1), 248–265.

Recalcati, C., Siena, M., Riva, M., Bollani, M., & Guadagnini, A. (2024). Stochastic assessment of dissolution at fluid-mineral interfaces. *Geophysical Research Letters*, *51*(7), e2023GL108080.

Scholz, C. H. (2019). *The mechanics of earthquakes and faulting*. Cambridge university press.

Seigneur, N., Mayer, K. U., & Steefel, C. I. (2019). Reactive transport in evolving porous media. *Reviews in Mineralogy and Geochemistry*, *85*(1), 197–238.

Stauffer, F., Bayer, P., Blum, P., Molina-Giraldo, N., & Kinzelbach, W. (2014). *Thermal use of shallow groundwater*.

---

## Author Comment (AC2)

**Referee #2**

In what follows, we respond in turn to each of the comments (included verbatim, in **bold**).

**Present work tries to develop a semi-analytical solution for the so-called reactive Lauwerier Problem (a thermos-hydro-chemical exercise with several simplification assumptions) aiming to evaluate the related porosity evolutions of the confined rock. The manuscript presents the development of a method that integrates various ideas, assumptions, and approaches, many of which have been previously presented. I believe that the manuscript is much better suited to possible publication as a 'Method' or 'Short Communications' paper than a 'Research Article'.**

We respectfully disagree with the assessment provided by the Referee suggesting that the contribution of the present paper only "integrates various ideas, assumptions, and approaches that have been previously presented" and welcome direction to any literature we may have overlooked. First and foremost, our work represents an original, analytical contribution. To the best of our knowledge, there are no existing previously published analytical solutions for thermally-driven reactive flow in this type of geometry, i.e. injection of thermal fluid from a point source into a confined aquifer. In 1955, Lauwerier derived an analytical solution for the injection of thermal fluid into a confined reservoir, famously known as "the Lauwerier Problem" [Lauwerier, 1955]. Now, 70 years later, we have leveraged Lawrier's solution to describe temperature-dependent solubility and develop a thermally-driven reactive transport solution. We have termed this problem "The Reactive Lauwerier Problem" in our work. Our contribiution predicts the coupling among temperature, solute concentration, flow rate, and porosity providing a novel understanding of thermally-driven reactive flow systems, which are relevant to natural, geothermal, and $CO_2$ applications.

Secondly, we emphasize that the solutions presented are fully analytical, without any associated approximations that would classify them as "semi-analytical". Furthermore, the manuscript presents also the solutions for the evolution of solute profiles in the aquifer (Eqs. 15 and 20), not solely porosity evolution (Eqs. 17 and 21). Additionally, we both present the solutions and use them to investigate common important case studies, including (I) hot $CO_2$-rich water injection into carbonate aquifers and (II) hot silica-rich water injection, leading to mineral dissolution and precipitation, respectively.

Lastly, we argue the derivation of analytical solutions is more than a technical advancement or exercise. Innovative analytical solutions are key to developing a basic theoretical description of physical processes, offer an understanding of the underlying mechanisms of a problem, and reveal relationships between variables and the fundamental properties of the system. As described in the Aims & Scope section, HESS encourages submissions of both fundamental and theoretical research. Therefore, we believe our work, which involves both theoretical and important case study investigations, is well-suited for publication as a Research Article in HESS.

1. **The primary question would be the effectiveness of the method. There is no validation for the implementation of the methodology presented in this work (it is clear that I am not taking about validation of the assumption t' ≈ t).**

We do not fully understand what the Reviewer means when referring to the effectiveness of the method. The article presents analytical solutions, not a numerical method, numerical computation is used in few places only to prevent the integer overflow, when one needs to deal with exponentiation of very large negative numbers. We are happy to address the Reviewer's comment with additional insight.

2. **Authors may decide to convert and perform analysis in an dimensionless conditions.**

As the Referee noted, in this work, we have chosen dimensional presentation over dimensionless presentation. This decision is based on the investigation of case studies presented in Section 3, which requires the use of explicit parameters with prescribed values. Consequently, we believe that dimensional presentation is more suitable in this case, in order to facilitate comparison with real-world case studies.

3. **Please clearly state the programing environment that Authors scripted. Do Authors develop a toolkit to perform simulations or use an available commercial/open-source software?**

Following the Reviewer's comment, the manuscript will be revised to state that Matlab was the programming software used. However, let us reiterate again that almost all of the curves presented in the manuscript represent analytical, closed-form solutions, which can be plotted using any graphical software. In specific instances, an approximate numerical solution was used to avoid integer overflow (Eq. D.2 in Appendix D). The manuscript currently states clearly when Eq. D.2 is used (lines 378, 395, and 496). In particular, to obtain the porosity profiles (Figs. 2b-c and 3) at specific times, an iterative numerical solution of Eq. D.2 was employed to avoid integer overflow. The manuscript will be revised to state more precisely when this iterative numerical procedure is used to attain the porosity profiles.

4. **Authors suggested that present work can stand as a benchmark for complex coupled numerical codes. Then I recommend making the Code & Data availability to "Publicly Available" (not on request).**

We agree with the Referee that the code and data should be publicly available, and therefore, we will make all code and data publicly accessible (and not just to Referees). However, it should be noted that the solutions (Eqs. 15, 17, D.2, 20, and 21) can be easily implemented in short scripts.

5. **It is hard to follow the storytelling of the manuscript. I expect that the manuscript (by itself) must be enough to understand "What the Authors did?", instead supplementary materials can support some details on "How they technically implement the analysis?". Indeed, I see that the provided information is mixed among the manuscript and supplementary materials. I would strongly recommend Authors to rearrange the structure of the work.**

We agree with the Referee that the manuscript itself should be sufficient to understand the work, with supplementary materials supporting technical details. This is also the philosophy that we were trying to follow when writing this manuscript. We kindly ask the Referee for specific comments on where it would be beneficial to move material from the Appendix to the main text. In our view, overloading the text with derivations could disrupt the flow of the manuscript, but we are open improving clarity in certain sections that may lack sufficient detail.

This work includes supportive appendices A-E. Their precise functionality is described below,

    a)   Appendix A: An Extended Form of the Conservation Equations

This appendix provides the expanded versions of the conservation equations and the assumptions that lead to the final equations appearing in the main text (section 2.3) and used to develop the solutions. Consequently, it supports the main work and is appropriate to appear in its place.

b)   Appendix B: Timescales Analysis to Validate the Quasi-static Assumption

This appendix provides an in-depth analysis and discussion related to the quasi-static assumption. This section also supports the main work, and adding it to the main text would make the manuscript unnecessarily long and tedious. To make this section clearer, it will be revised to include separate equations instead of inline equations.

c)   Appendix C: Lauwerier Solution Validity Assuming $t' \approx t$

This appendix provide support to the main text by showing the validity of the assumption $t' \approx t$.

d)   Appendix D: Asymptotic Expansion for the Disequilibrium Solutions

This appendix presents an approximation for Eqs. 15 and 20, which are useful for efficient computation and to avoid computational overflow. Therefore, it should not be part of the main text.

e)   Appendix E: Permeability of an Aquifer with Nonuniform Porosity Profile

This section provides technical details for calculation of the effective permeability and hence appears as an appendix.

**6.   - Line 152: I cannot see "numerical simulations" in present work, please specify.**

Please see reply to comment 8.

**7.   - Please avoid discussion of the results of the Figures in captions (instead of the text body).**

All the interpretations that appear in the caption also appear in the main text. We believe that non-purely technical captions are more informative to the reader. However, we are open to revising them.

**8.   - Line 190: last sentence "Thermal variations in the…" is incomplete.**

Will be corrected.

**9.   - Table 1: Please move it to the end of paper, as "Nomenclature".**

We placed Table 1 early in the manuscript for easy reference to the list of parameters, but we are open to revise the location of Table 1.

**10. - Please rearrange Eq. (4) not to start with zero value.**

The presentation of the equation in its current form emphasizes the transient term is equal to zero. Typically, transient terms appear on the left-hand-side of the equation (see e.g., Battiato et al., 2009; Steefel et al., 2005; Steefel & Maher, 2009). Staying consistent with the presentation in past literature, the current presentation emphasizes that this is a steady-state equation, and the study adopts a quasi-static approach.

**11. - Please avoid discussion of the literature in supplementary materials.**

Please refer to the reply to comment 5.

**References**

Battiato, I., Tartakovsky, D. M., Tartakovsky, A. M., & Scheibe, T. D. (2009). On breakdown of macroscopic models of mixing-controlled heterogeneous reactions in porous media. *Adv. Water Resour.*, *32*(11), 1664–1673.

Lauwerier, H. (1955). The transport of heat in an oil layer caused by the injection of hot fluid. *Applied Scientific Research, Section A*, *5*(2), 145–150.

Steefel, C., Depaolo, D., & Lichtner, P. (2005). Reactive transport modeling: An essential tool and a new research approach for the Earth sciences. *Earth and Planetary Science Letters*, *240*, 539–558. https://doi.org/10.1016/j.epsl.2005.09.017

Steefel, C. I., & Maher, K. (2009). Fluid-Rock Interaction: A Reactive Transport Approach. *Reviews in Mineralogy and Geochemistry*, *70*(1988), 485–532. https://doi.org/10.2138/rmg.2009.70.11

---

## Author Response (AR1)

**Reply letter to the Reviewers' comments: hess-2023-307 –** Solutions and Case Studies for Thermally-driven Reactive Transport and Porosity Evolution in Geothermal Systems ("Reactive Lauwerier Problem")

**Dear Editor,**

We thank the Reviewers for their comments, which have significantly helped improve the manuscript.

Particularly, we wish to note that this work represents an original, analytical contribution: in 1955, Lauwerier derived an analytical solution for the injection of thermal fluid into a confined reservoir, famously known as "the Lauwerier Problem" [Lauwerier, 1955]. In this study, 70 years later, we have leveraged Lauwerier's solution to describe temperature-dependent solubility and develop a thermally-driven reactive transport solution (termed here as "The Reactive Lauwerier Problem"). The solutions predicts the coupling among temperature, solute concentration, flow rate, and porosity providing a novel understanding of thermally-driven reactive flow systems, which are relevant to natural, geothermal, water resources and $CO_2$ applications. Additionally, we both present the solutions and use them to investigate common important case studies, including (I) hot $CO_2$-rich water injection into carbonate aquifers and (II) hot silica-rich water injection, leading to mineral dissolution and precipitation, respectively.

Lastly, we wish to emphasize that the value of analytical solutions extends beyond technical advancements. These solutions are key to developing a basic theoretical description of physical processes, understanding the underlying mechanisms of a problem, and revealing relationships between variables and the fundamental properties of the system. As described in the Aims & Scope section, HESS encourages submissions of both fundamental and theoretical research. Therefore, we believe our work, which involves both theoretical developments and important case study investigations, is well-suited for publication as a Research Article in HESS.

Below we provide our detailed itemized response to all comments (included verbatim, in **bold**). This letter is supplemented by a PDF highlighting changes to the text vs. the previous submission.

**Referee #1**

We thank the Reviewer for the careful review, which helped us to improve the manuscript. We are glad the Reviewer found the manuscript to be well-written, well-organized, and noted the topic is of high interest to the scientific community.

1. **I think the Introduction is a bit wordy and can be summarized in 4-5 concise paragraphs raising the main points.**

Following the Referee's comment, the introduction was shortened while retaining important background information and necessary details for the presentation (with a total of 1400 words compared to 1900 words in the previous version).

Important issues now addressed more concisely in the introduction include: a) a description of the natural phenomenon investigated and its relation to subsurface processes; b) definitions of timescales and the thermal and solute Péclet numbers governing the problem; c) the coupled Thermo-Hydro-Chemical (THC)

processes and their applications; d) a review of common thermally-driven geochemical reactions, focusing on sandstones and carbonates that are later included in the case studies; e) a concise discussion on the applicability and rigorous testing of THC numerical codes; f) a concise review of existing THC analytical solutions; and lastly, g) an outline of the paper. We believe that the current version of the introduction provides all the necessary information and settings needed for the study in a more concise manner.

2. **Considering the main model assumptions, how applicable/reliable is constant fluid density with temperature changes, in particular, for CO2 as the working fluid? The same question will be raised for the assumption of having the same heat capacity for both confining rocks and the aquifer.**

We thank the Reviewer for noting these issues. The Lauwerier solution (Lauwerier, 1955) refers to a thin confined aquifer layer, in which no substantial vertical temperature gradients develop (lines 173-176).

Regarding the potential effect of variable density on lateral flow, changes in density due to heating or cooling along the flow path can cause variations in flow rate. Specifically, fluid contraction during cooling can lead to a decrease in flow rate, while fluid expansion during heating can lead to an increase in flow rate along the path. However, in most cases, the overall temperature change does not exceed, e.g., $\Delta T =$ 80 °C, resulting in a low change in water density, typically around 2%. Consequently, this change does not have a substantial effect on flow and reactive transport (assuming there is no phase change). However, for the case of supercritical $CO_2$, the changes in density can be much larger, and in some cases the incompressibility assumption may not be appropriate. The manuscript was revised to discuss this issue (lines 180-182).

Regarding the uniform heat capacities, the reactive Lauwerier solution, like the original Lauwerier solution, does not inherently assume uniform heat capacities. This assumption is made here for simplicity and to avoid cumbersome equations. By adopting alternative definitions of the parameters in Eqs. 10 and 18, non-uniform heat capacities of the confining layers and the aquifer can be considered in reactive Lauwerier solution (refer to Eqs. 3.122 and 3.131 and associated definitions in Stauffer et al. (2014)). The manuscript was revised to make this important issue clear (lines 256-259).

3. **The authors have used a power-law relationship between the porosity and permeability of the aquifer to calculate the effective permeability. The authors have raised the limited predictive capability of these types of relations as well and mentioned that they solely used this type of relations to evaluate the general trends. Although this type of relationship could help evaluate the general trend of permeability evolution in mineral dissolution cases, they may not be the best choice for anti-correlated porosity-permeability changes in the systems when the percolating fluid has a weak capacity for dissolution or when we have mineral precipitation. Why did the authors not use other types of porosity-permeability relationships (e.g., a two-parameter exponential model) for sandstone aquifers where they expected to see mineral precipitation?**

We agree with the Reviewer that for specific cases and reactive flow duration (e.g., Garing et al., 2015) there may be more appropriate choices for porosity-permeability relations. However, we note this falls outside the scope of the current work. Here, we emphasize the different trends of aquifer permeability evolution due to dissolution in a carbonate aquifer and silica precipitation. Consequently, we selected a

generic form of the porosity-permeability relation. The power-law relation demonstrates that permeability will typically rise within relatively short geological timescales in carbonate aquifers due to dissolution while substantial permeability changes due to silica precipitation occur on the order of tens of thousands of years. We argue, besides unique cases, the conclusions will remain valid regardless of the porosity-permeability relation used. Following the Reviewer comment, the discussion in the manuscript was revised to better refer to this issue (lines 453-457).

**4. It seems that the last sentence in the Figure 1 caption is not complete. Please review the caption and correct it.**

Thank you for catching this oversight, the caption for Figure 1 has been corrected.

**5. How reliable are the utilized values for the surface area of the carbonates and sandstones? Are they within the reported range for the surface area of these rocks in the literature?**

The values for the specific surface area of the sandstones utilized in the present work (i.e., $A_s = 10^4 \, m^{-1}$) fall within the ranges reported in the literature for common sandstone formations ($10^4$-$10^5 \, m^{-1}$; Hussaini & Dvorkin, 2021; Lai et al., 2015). However, it is important to note the ongoing research and discussions regarding the uncertainties in the estimation of reactive surface area in various rocks and under different reactive transport conditions (Noiriel and Daval, 2017; Recalcati et al., 2024; Seigneur et al., 2019). To address the Reviewer's comment, the range of values reported in the literature including additional references has been added in the revised manuscript.

For fractured carbonates, previous studies have demonstrated that in cases of large permeability contrast between fractures and the rock matrix, only the fracture surface area effectively participates in the reaction (i.e., constitutes the reactive surface area). This has been demonstrated by field case studies (e.g., MacQuarrie & Mayer, 2005; Pacheco & Alencoão, 2006). In this case, the reactive surface area can be calculated using $A_s = 2 \cdot \kappa \cdot RF$ where $\kappa$ is fracture density (defined as the number of fractures per unit volume), the factor of 2 accounts for the presence of two surfaces, and RF is the roughness factor (see Deng et al., 2018). Assuming $\kappa = 1/3^3 \, m^{-3}$ and RF = 1.35, results in As = 0.1 m-1. Typical values of $\kappa$ and fracture spacing can span a substantial range and may be higher or lower (Narr and Suppe, 1991; Scholz, 2019). Following the Reviewer's comment, this issue was further clarified and references were added to the revised manuscript (lines 337-341).

**Referee #2**

**Present work tries to develop a semi-analytical solution for the so-called reactive Lauwerier Problem (a thermos-hydro-chemical exercise with several simplification assumptions) aiming to evaluate the related porosity evolutions of the confined rock. The manuscript presents the development of a method that integrates various ideas, assumptions, and approaches, many of which have been previously presented. I believe that the manuscript is much better suited to possible publication as a 'Method' or 'Short Communications' paper than a 'Research Article'.**

We respectfully disagree with the assessment provided by the Referee suggesting that the contribution of the present paper only "integrates various ideas, assumptions, and approaches that have been previously

presented" and welcome direction to any literature we may have overlooked. First and foremost, our work represents an original, analytical contribution. To the best of our knowledge, there are no existing previously published analytical solutions for thermally-driven reactive flow in this specific geometry. In 1955, Lauwerier derived an analytical solution for the injection of thermal fluid into a confined reservoir, famously known as "the Lauwerier Problem" [Lauwerier, 1955]. Now, 70 years later, we have leveraged Lawrier's solution to describe temperature-dependent solubility and develop a thermally-driven reactive transport solution. We have termed this problem "The Reactive Lauwerier Problem" in our work. Our contribiution predicts the coupling among temperature, solute concentration, flow rate, and porosity providing a novel understanding of thermally-driven reactive flow systems, which are relevant to natural, geothermal, and $CO_2$ applications.

Secondly, we emphasize that the solutions presented are fully analytical, without any associated approximations that would classify them as "semi-analytical". Furthermore, the manuscript presents also the solutions for the evolution of solute profiles in the aquifer (Eqs. 15 and 20), not solely porosity evolution (Eqs. 17 and 21). Additionally, we both present the solutions and use them to investigate common important case studies, including (I) hot $CO_2$-rich water injection into carbonate aquifers and (II) hot silica-rich water injection, leading to mineral dissolution and precipitation, respectively. To emphasize this last issue the manuscript title was revised.

Lastly, we argue that the derivation of analytical solutions is more than a technical advancement or exercise. Innovative analytical solutions are key to developing a basic theoretical description of physical processes, offer an understanding of the underlying mechanisms of a problem, and reveal relationships between variables and the fundamental properties of the system. As described in the Aims & Scope section, HESS encourages submissions of both fundamental and theoretical research. Therefore, we believe our work, which involves both theoretical and important case study investigations, is well-suited for publication as a Research Article in HESS.

6. **The primary question would be the effectiveness of the method. There is no validation for the implementation of the methodology presented in this work (it is clear that I am not taking about validation of the assumption t' ≈ t).**

We do not fully understand what the Reviewer means when referring to the effectiveness of the method. The article presents analytical solutions, not a numerical method, numerical computation is used in few places only to prevent the integer overflow, when one needs to to deal with exponentiation of very large numbers. We are happy to address the Reviewer's comment with additional insight.

7. **Authors may decide to convert and perform analysis in an dimensionless conditions.**

As the Referee noted, in this work, we have chosen dimensional presentation over dimensionless presentation. This decision is based on the investigation of case studies presented in Section 3, which requires the use of explicit parameters with prescribed values. Consequently, we believe that dimensional presentation is more suitable in this case to facilitate comparison with real-world case studies.

8. **Please clearly state the programing environment that Authors scripted. Do Authors develop a toolkit to perform simulations or use an available commercial/open-source software?**

Following the Reviewer's comment, the manuscript was revised to state that MATLAB was the programming software used (lines 347-348). However, let us reiterate again that almost all of the curves presented in the manuscript represent analytical, closed-form solutions, which can be plotted using any coding software. In specific instances, an approximate numerical solution was used to avoid integer overflow (Eq. D.2 in Appendix D). The manuscript currently states clearly when Eq. D.2 is used (Appendix D, lines 348-349, 352 and 604-605). In particular, to obtain the porosity profiles (Figs. 2b-c and 3) at later times ($t$ = 100 kyr), an iterative numerical solution of Eq. D.2 was employed. The manuscript was revised to state more precisely when this iterative numerical procedure is used to attain the porosity profiles (Appendix D).

9. **Authors suggested that present work can stand as a benchmark for complex coupled numerical codes. Then I recommend making the Code & Data availability to "Publicly Available" (not on request).**

We agree with the Referee that the code and data should be publicly available, and therefore, we made all code and data publicly accessible (and not just to Referees). However, it should be noted that the solutions (Eqs. 15, 17, D.2, 20, 21 and D.3) can be easily implemented in short scripts.

10. **It is hard to follow the storytelling of the manuscript. I expect that the manuscript (by itself) must be enough to understand "What the Authors did?", instead supplementary materials can support some details on "How they technically implement the analysis?". Indeed, I see that the provided information is mixed among the manuscript and supplementary materials. I would strongly recommend Authors to rearrange the structure of the work.**

We agree with the Referee that the manuscript itself should be sufficient to understand the work, with supplementary materials supporting technical details. This is also the philosophy that we were trying to follow when writing this manuscript. We kindly ask the Referee for specific comments on where it would be beneficial to move material from the Appendix to the main text. In our view, overloading the text with derivations could disrupt the flow of the manuscript, but we are open improving clarity in certain sections that may lack sufficient detail.

This work includes supportive appendices A-E. Their precise functionality is described below,

a) Appendix A: An Extended Form of the Conservation Equations

This appendix provides the expanded versions of the conservation equations and the assumptions that lead to the final equations appearing in the main text (section 2.3) and used to develop the solutions. Consequently, it supports the main work and is appropriate to appear in its place.

b) Appendix B: Timescales Analysis to Validate the Quasi-static Assumption

This appendix provides an in-depth analysis and discussion related to the quasi-static assumption. This section also supports the main work, and adding it to the main text would make the manuscript unnecessarily long and tedious. To make this section clearer, it was revised to include separate equations instead of inline equations (Eq. B.1).

c) Appendix C: Lauwerier Solution Validity Assuming $t' \approx t$

This appendix provide support to the main text by showing the validity of the assumption $t' \approx t$.

d) Appendix D: Asymptotic Expansion for the Disequilibrium Solutions

This appendix presents an approximation for Eqs. 15 and 20, which is useful for efficient computation and to avoid computational overflow. Therefore, it should not be part of the main text.

e) Appendix E: Permeability of an Aquifer with Nonuniform Porosity Profile

This section provides technical details for calculation of the effective permeability and hence appears as an appendix.

**11. - Line 152: I cannot see "numerical simulations" in present work, please specify.**

Following the Referee comment, the manuscript was revised (lines 119-120). Please also see reply to comment 8.

**12. - Please avoid discussion of the results of the Figures in captions (instead of the text body).**

All the interpretations that appear in the caption also appear in the main text. We believe that non-purely technical captions are more informative to the reader. However, we are open to revising them based on specific feedback.

**13. - Line 190: last sentence "Thermal variations in the..." is incomplete.**

Revised.

**14. - Table 1: Please move it to the end of paper, as "Nomenclature".**

We placed Table 1 early in the manuscript for easy reference to the list of parameters, but we are open to revise the location of Table 1.

**15. - Please rearrange Eq. (4) not to start with zero value.**

The presentation of the equation in its current form emphasizes the transient term is equal to zero. Typically, transient terms appear on the left-hand-side of the equation (see e.g., Battiato et al., 2009; Steefel et al., 2005; Steefel & Maher, 2009). Staying consistent with the presentation in past literature, the current presentation emphasizes that this is a steady-state equation, and the study adopts a quasi-static approach.

**16. - Please avoid discussion of the literature in supplementary materials.**

Please refer to the reply to comment 5.

**References**

Battiato, I., Tartakovsky, D. M., Tartakovsky, A. M., and Scheibe, T. D.: On breakdown of macroscopic models of mixing-controlled heterogeneous reactions in porous media, Adv Water Resour, 32, 1664–1673, 2009.

Deng, H., Molins, S., Trebotich, D., Steefel, C., and DePaolo, D.: Pore-scale numerical investigation of the impacts of surface roughness: Upscaling of reaction rates in rough fractures, Geochim. Cosmochim. Acta, 239, 374–389, 2018.

Garing, C., Gouze, P., Kassab, M., Riva, M., and Guadagnini, A.: Anti-correlated porosity–permeability changes during the dissolution of carbonate rocks: experimental evidences and modeling, Transp. Porous Media, 107, 595–621, 2015.

Hussaini, S. R. and Dvorkin, J.: Specific surface area versus porosity from digital images, J. Pet. Sci. Eng., 196, 107773, 2021.

Lai, P., Moulton, K., and Krevor, S.: Pore-scale heterogeneity in the mineral distribution and reactive surface area of porous rocks, Chem. Geol., 411, 260–273, 2015.

Lauwerier, H.: The transport of heat in an oil layer caused by the injection of hot fluid, Appl. Sci. Res. Sect. A, 5, 145–150, 1955.

MacQuarrie, K. T. and Mayer, K. U.: Reactive transport modeling in fractured rock: A state-of-the-science review, Earth-Sci. Rev., 72, 189–227, 2005.

Narr, W. and Suppe, J.: Joint spacing in sedimentary rocks, J. Struct. Geol., 13, 1037–1048, 1991.

Noiriel, C. and Daval, D.: Pore-scale geochemical reactivity associated with CO2 storage: New frontiers at the fluid–solid interface, Acc. Chem. Res., 50, 759–768, 2017.

Pacheco, F. A. L. and Alencoão, A. M. P.: Role of fractures in weathering of solid rocks: narrowing the gap between laboratory and field weathering rates, J. Hydrol., 316, 248–265, 2006.

Recalcati, C., Siena, M., Riva, M., Bollani, M., and Guadagnini, A.: Stochastic assessment of dissolution at fluid-mineral interfaces, Geophys. Res. Lett., 51, e2023GL108080, 2024.

Scholz, C. H.: The mechanics of earthquakes and faulting, Cambridge university press, 2019.

Seigneur, N., Mayer, K. U., and Steefel, C. I.: Reactive transport in evolving porous media, Rev. Mineral. Geochem., 85, 197–238, 2019.

Stauffer, F., Bayer, P., Blum, P., Molina-Giraldo, N., and Kinzelbach, W.: Thermal use of shallow groundwater, 2014.

Steefel, C., Depaolo, D., and Lichtner, P.: Reactive transport modeling: An essential tool and a new research approach for the Earth sciences, Earth Planet. Sci. Lett., 240, 539–558, https://doi.org/10.1016/j.epsl.2005.09.017, 2005.

Steefel, C. I. and Maher, K.: Fluid-Rock Interaction: A Reactive Transport Approach, Rev. Mineral. Geochem., 70, 485–532, https://doi.org/10.2138/rmg.2009.70.11, 2009.